



**Petrographic and Petrophysical Characteristics of Lower Cretaceous**
**Sandstones from northern Israel, determined by micro-CT imaging and**
**analytical techniques**
Peleg Haruzi[1], Regina Katsman[1], Baruch Spiro[1,2], Matthias Halisch[3] and Nicolas Waldmann[1]
[1] The Dr. Moses Strauss Department of Marine Geosciences, Faculty of Science and Science Education, The
University of Haifa, Haifa, Mount Carmel 3498838, Israel
[2] Department of Earth Sciences, Natural History Museum, Cromwell Road, London SW7 5BD, UK
[3] Leibniz Institute for Applied Geophysics (LIAG), Dept. 5 – Petrophysics & Borehole Geophysics, Stilleweg
2, D-30655 Hannover, Germany
*Correspondence to:* Regina Katsman (rkatsman@univ.haifa.ac.il)
**Keywords:** sandstone, petrography, petrophysics, micro-CT imaging, pore-scale modelling





**Abstract**

19        In this study petrophysical characteristics of three consecutive sandstone layers of the Lower Cretaceous

Hatira Formation from northern Israel were comprehensively investigated and analysed. The methods used
were: experimental petrographic and petrophysical methods, 3D micro-CT imaging and pore-scale single-
phase flow modelling, conducted in parallel. All three studied sandstone layers show features indicative of
high textural and mineralogical maturity in agreement with those reported from the Kurnub Group in other
localities in the Levant. The occurrence of cross-bedding in layers enriched in silt and clay, between the quartz
arenite rich beds, may suggest a deposition in a fluvial environment. A higher degree of Fe-ox cementation
was observed in the top layer contrasting with a low extent of Fe-ox cementation in the bottom layer. Both
quartz-arenite layers are located above and below the intermediate 20 cm thick least permeable quartz wacke
sandstone layer. The latter presumably prevented the supply of the iron-rich meteoric water to the bottom
layer. Evaluated micro-scale geometrical rocks properties (pore size distribution, pore throat size,
characteristic (pore-throat) length, pore throat length of maximal conductance, specific surface area, grain
roughness) and macro-scale petrophysical properties (porosity and tortuosity) predetermined the permeability
of the studied layers. Large-scale laboratory porosity and permeability measurements show low variability in
the quartz arenite (top and bottom) layers, and high variability in the quartz wacke (intermediate) layer. These
degrees of variability are confirmed also by anisotropy and homogeneity analyses conducted in the µCT-
imaged geometry. Qualitative evaluation of anisotropy (based on statistical distribution of pore space) and
connectivity (using Euler Characteristic) were correlated with mineralogy and grain surface characteristics,
clay matrix and preferential location of cementation. Two scales of porosity variations were found with
variogram analysis of the upper quartz arenite layer: fluctuations at 300 µm scale due to pores size variability,
and at 2 mm scale due to the appearance of high and low porosity occlusion by ferruginous bands showing
iron oxide cementation. We suggest that this cementation is a result of iron solutes transported by infiltrating
water through preferential permeable paths in zones having large grains and pores. Fe-ox precipitated as a
result of reaction with oxygen in a partly-saturating realm at the large surface area localities adjacent to the
preferential conducting paths. The core part of the study is the investigation of macroscopic permeability,
upscaled from pore-scale velocity field, simulated by free-flow in real µCT-scanned geometry on mm-scale
sample. The results show an agreement with lab petrophysical estimates on cm-scale sample for the top and
bottom layers. Estimated permeability anisotropy correlates with the presence of beddings with 2 mm scale



variability in the top layer. The results show that this kind of anisotropy rather than a variability at the pore-scale controls the macroscopic rock permeability. Therefore, we suggest that in order to upscale reliably to the lab permeability, a sufficiently large modelling domain is required to capture the textural features that appear at a scale larger than the pore scale. We also discuss imaging and modelling practices able to preserve the characteristics of the pore network during the entire computational workflow procedure, applicable to studies in the fields of hydrology, petroleum geology, or sedimentary ore deposits.

.

# 1.  Introduction

## 1.1. Lower Cretaceous sandstone as a reservoir rock

Lower Cretaceous sandstone units serve as a reservoir rocks for hydrocarbons in various places over the world (e.g. Borgomato et al., 2013; Peksa et al., 2015; Akinlotan, 2016) including the largest clastic oilfield (Greater Burgan, Kuwait; Reynolds, 2017), and in Israel (e.g. the Heletz onshore and Yam offshore oil fields; Gardosh and Tannenbaum et al., 2011). Marine Lower Cretaceous Heletz units from Southern Israel have been comprehensively characterized (e.g. Calvo et al., 2011; Niemi et al., 2016; Tatomir et al., 2016) in a course of a pilot project on potential $CO_2$ storage in a deep saline reservoir site, in contrast to the non-marine Lower Cretaceous Hatira Formation units from the northern Israel, explored in our study.

Macroscopic effective rock properties (e.g. porosity and permeability) are usually evaluated from the conventional laboratory experiments that sometimes suffer from errors due to sample's local heterogeneity, their small quantity, or insufficient financial resources (e.g. Halisch, 2013). These macro-scale characteristics are predefined by micro-scale descriptors (Cerepi et al., 2002; Haoguang et al., 2014; Nelson, 2009) and thus can be obtained from their upscaling (e.g. Wildenschild and Sheppard, 2013; Andrä et al., 2013; Bogdanov et al., 2011; Narsilio et al., 2009).

Numerous attempts, which have been made in the past decades demonstrated that a pore-scale description provides useful details about the dynamics of fluids transfer and the chemical reactions in the porous media (e.g. Kalaydjan, 1990; Whitaker, 1986). As a result, pore-scale imaging and flow simulations (Bogdanov et al., 2012; Blunt et al., 2013; Cnudde et al., 2013; Wildenschild and Sheppard, 2013; Halisch, 2013) started to serve as a reliable method to characterize flow and rock properties at a pore-scale. The





advantages of these techniques are their non-destructive character and capability to provide a reliable 3D
information about the real pore-space.
This paper presents a case study of three consecutive sandstone layers of the Lower Cretaceous Hatira
Formation from Northern Israel. These are for the first time comprehensively investigated with experimental
petrographic and petrophysical methods, 3D micro-computed tomography (μ-CT) imaging and pore-scale
flow modelling, and statistical anisotropy analysis, conducted in parallel at different scales. As a core part of
the study, we link the micro-scale geometrical and topological rock properties and macro-scale permeability.
The statistically evaluated permeability anisotropy is found to correlate with the presence of bedding features
at a mm scale quantified in parallel by mineralogical, textural and grain surface analysis. We suggest that a
sufficiently large size of the modelling domain is required in order to upscale reliably to the lab scale
permeability, to capture the textural features that appear at a scale coarser than the pore scale. We also address
features of the depositional environments. We discuss imaging and modelling practices, aimed to preserve the
relevant characteristics of the pore network during the entire computational workflow, applicable to studies in
the fields hydrology, petroleum geology, or sedimentary ore deposits.
Detailed characterization of the non-marine units of Hatira Fm. from the northern part of Israel
conducted in our study may have a wider significance. The information derived from the measurements should
allow the improvement in the identification of sedimentation patterns and evaluation of depositional, climatic,
tectonic and eustatic conditions at the Lower Cretaceous sections in this and other locations: e.g. in Europe
(Akinlotan, 2017), China (Li et al., 2016) and South America (Ferreira et al., 2016).

**1.2. Geological setting**
The study is based on samples collected from an outcrop at Wadi E'Shatr near Ein Kinya on the southern
slopes of Mt. Hermon (WGS84 Long. 33.239118, Lat. 35.741117, Alt. 924 m), Fig. 1. The outcrop consists
of sandstones of the Lower Cretaceous Hatira Fm. (Sneh and Weinberger, 2003). The Hatira formation acts
as reservoir rock for hydrocarbons in Israel (Fig 1a): on shore; Heletz (Grader and Reiss, 1958; Grader, 1959;
Shenhav 1971), and off-shore; Yam Yaffo (Gardosh and Tannenbaum 2014) (Cohen, 1971; Cohen, 1983;
Calvo, 1992; Calvo et al., 2011).




The Hatira Formation is the lower part of the Kurnub Group of Lower Cretaceous Neocomian –
Barremian age. The term is used in Israel and Jordan and is equivalent to Grès de Base in Lebanon (Massad,
1976). It occurs in Israel in outcrops from the Eilat area along the rift valley, in the central Negev with the
northernmost outcrops on Mount Hermon. It forms a part of a large Palaeozoic –Mesozoic platform and
continental margin deposits in north east Africa and Arabia. It consists of siliciclastic units typically dominated
by quartz rich sandstones (Kolodner et al., 2009 and references therein). The Underlying Palaeozoic
sandstones cover large areas in North Africa and Arabia from Morocco to Oman. These overly a Precambrian
basement affected by the Neoproterozoic (pan African) orogenesis (Klitsch, 1981; Garfunkel, 1988, 1999;
Avigad et al., 2003, 2005). The lower Palaeozoic sandstones in Israel and Jordan originated from the erosion
of that Neoproterozoic basement, Arabian Nubian Shield, with contribution from older sources. The Lower
Palaeozoic sandstones (Cambrian and Ordovician) are described as first cycle sediments (Weissbrod and
Nachmias, 1986; Amireh, 1997; Avigad et al., 2005). Exposures of the Hatira Formation in the Central Negev,
the Arava Valley Eilat and Sinai were originally defined as the Wadi (Kurnub) Hatira Sandstone (Shaw 1947).
The largely siliciclastic section of the Hatira Fm. is intercalated with carbonates and shales representing marine
ingressions, increasing towards the north (Weissbrod, 2002).
The Lower Cretaceous sandstones of the Kurnub Group are described as super mature, cross-bedded
medium to fine grained, moderately sorted to well sorted, quartz arenites with a high ZTR index (Kolodner,
2009). The Zircon Tourmaline Rutile (ZTR) index of sandstones (Hubert, 1962) - is a measure of their
mechanical and chemical stability, with high values indicating a long history of transport and also an exposure
to aquatic environment (Hubert, 1962). The age spectrum of detrital zircon in the Lower Cretaceous Hatira
Fm. is dominated by Neoproterozoic age (0.55 to 0.65Ga) with various amounts of older Pre-Neoproterozoic,
spanning the range 0.95-2.65Ga. The similarity of the age spectra to those recorded from Cambrian and
Ordovician sandstone sections in Israel and Jordan (Kolodner, 2009), led to the conclusion that the lower
Cretaceous sandstones are mainly products of recycling of older siliciclastic rocks throughout the Phanerozoic.
This conclusion based on U/Pb chronology of zircons, reinforces earlier observations of that unit indicating
relatively scarce occurrence of siltstones and claystones in comparison to sandstones (Massad, 1976; Abed,
1982; Amireh, 1997). A petrographic evidence of recycling is the smooth surface of grains and even their
earlier overgrowths, ascribed to erosion of the first generation sandstone cement (Kolodner et al., 2009). The




sand was first eroded from the surface of the pan African orogeny ca. 400 Ma prior to its deposition in the
Lower Cretaceous sediments (Kolodner et al., 2009).
The Mount Hermon block from which the samples of the present study originate, was located at the
southern border of the Tethys Ocean during the Early Cretaceous (Bachman and Hirsch, 2006). The
palaeogeographical reconstruction indicates that the sandy Hatira Fm. (Fig. 1b) was deposited in a large basin,
which included both terrestrial and coastal environments such as swamps and lagoons (Sneh and Weinberger,
2003). The Hermon block located next to the Dead Sea Transform, was rapidly uplifted during the Neogene
(Shimron, 1998). The area is marked by intense erosion which resulted in extensive outcrops such as those
near Ein Kinya on the SE side of Wadi Esh Shatr. Saltzman (1967) described the Sandstones as Lower
Cretaceous – Aptian (L.C.1) referring to them as the Esh Shatr Formation. The Esh Shatr Formation overlies
with an angular unconformity the Jurassic Banias Basalts. It is overlain by the Ein El Assad limestone (L.C.2).
Sneh and Weinberger (2004) describe the Kurnub Group of Lower Cretaceous Neocomian-Barremian
age in the study area (Fig. 1d) as consisting of a volcanic sequence at the base, overlain uncomformably by
sandstone and clay layers of the Hatira Formation, with the upper unit of limestone marl chalk – the Nabi Said
Fm.
At the location of the section of Saltzman (1967) which is ca. 100 m SW away from the sampling area
for the present study, the 58 m thick variegated sandstone is interbedded with layers of clay and of clay-marl.
The sandy component is white-yellowish-brown/red consisting of largely angular, poorly sorted quartz grains
0.5 to 5 mm in diameter. On exposed surfaces the sandstone is hardened by iron oxides and perhaps calcareous
cement. Fresh exposures are however brittle. The outcrops show lenticular benches 0.2-1.0 m thick. The
bedding is generally normal-horizontal, and locally inclined or cross bedding. The clay rich interlayers are
grey and normally siltic and brittle. Locally these layers contain lignite.
The underlying volcanic sequence is 50-200 m thick (Shimron and Peltz, 1993). Analyses of the
underlying Banias basalt (Alkali Basalt from E'Shatr spring, Basanite flows at the E'Shatr Pass and E'shatr
spring) gave K-Ar ages of 108.2, 122.4 and 133.6 Ma respectively, expressed as assumed ages of 125Ma
(Wilson et al., 2000), i.e. Lower Cretaceous.



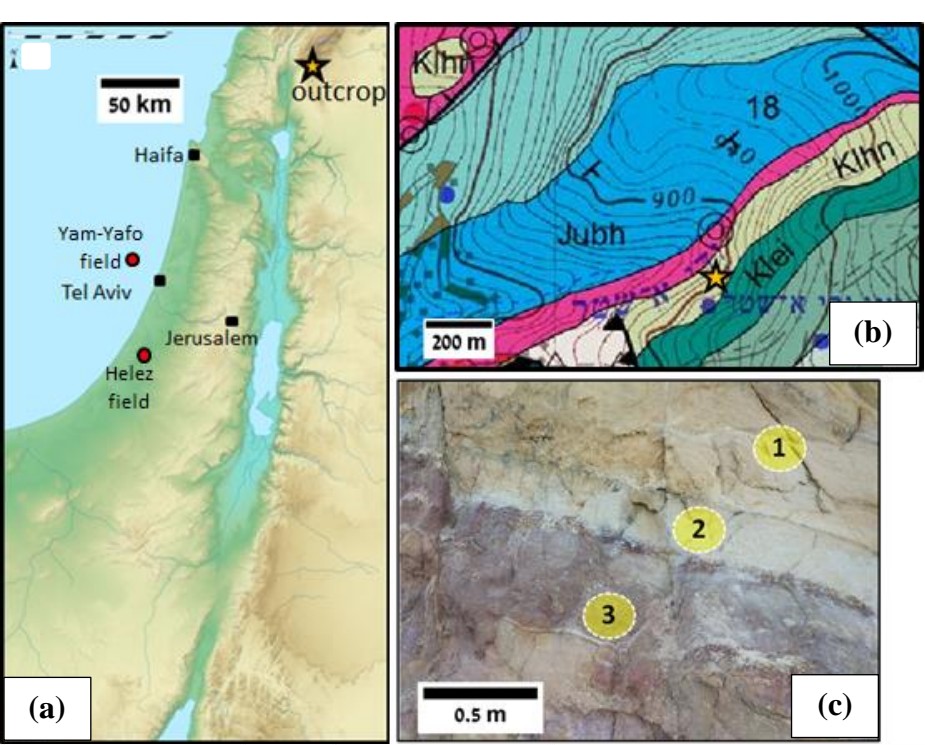

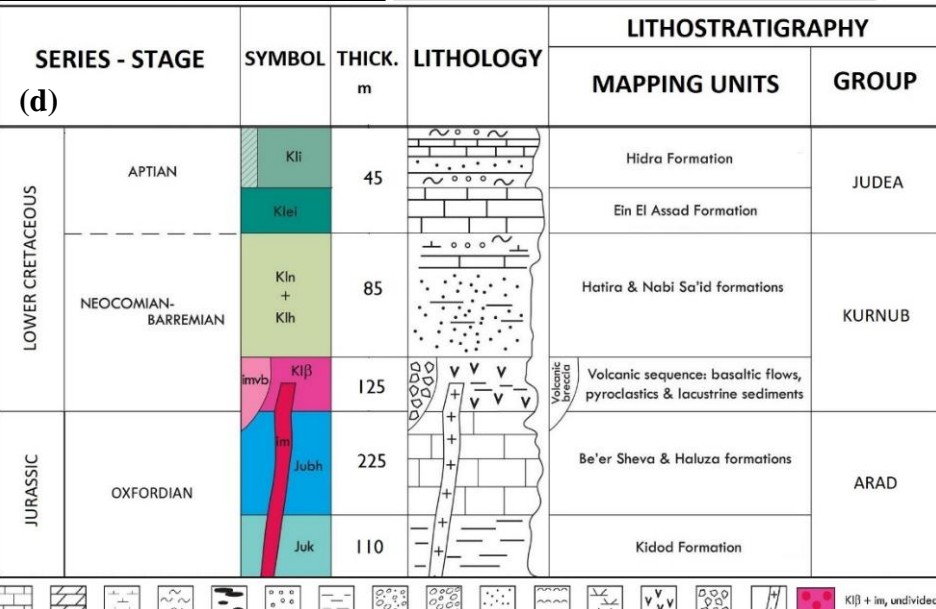



***Figure 1.*** *Geographical and Geological settings. **(a)** Schematic relief map of Israel: The site of Ein Kinya on*
*the Southern flanks of Mt. Hermon is indicated by a star. The map is modified from www.mapsland.com. **(b)***





*Geological map of Ein Kinya (WGS84 Long. 33.239118, Lat. 35.741117). The Hatira formation sandstone,*
*and the overlying limestone and marl Nabi Said Formation are marked as klhn. The outcrop where the samples*
*were retrieved is located on the southern slope of Wadi Al-Shattar hillside facing NW. The map is adopted*
*from Sneh and Weinberger (2014). (c) View of part of the outcrop of sandstones of the Lower Cretaceous*
*Hatira Formation at Ein Kinya, showing the layers from which samples were retrieved. These have distinct*
*colours – yellow-brown (1), grey-green (2), and red-purple (3). (d) Stratigraphic table of the geological map*
*(modified from Sneh and Weinberger, 2014).*
**2.    Methods**
**2.1. Samples description**
Samples were extracted from three consecutive layers of different colours that compose the stratigraphic
sequence (Figs. 1c, 1d). The lower layer (3) is ~1.5 m thick, composed of light (pale) red-purple in colour
sandstone with undulating bedding planes between the sub-layers. The middle layer (2) is 20 cm thick grey –
green shaly sandstone with dark horizons at the bottom and top. The upper layer (1) is 1.5m thick homogenous
brown- yellow sandstone. Large samples were retrieved in the field from these three layers noting the direction
perpendicular to the bedding planes (defined as z-direction in our study). Subsequently in the laboratory,
smaller sub-samples were prepared from these large samples for textural observations and various analytical
measurements.
**2.2. Laboratory and computational methods for rock characterization**
An integrated analytical program designed in our study used the following laboratory measurements and
computations conducted at different scales (from the core-scale reflecting the scale of the layers at the outcrop,
to the micro-scale reflecting the scale of the separate pores and grains) to comprehensively evaluate the
petrographic and petrophysical properties of the rock (Table 1). Specimens of a few cm-size were investigated
by petrographical and petrophysical lab methods. Specimens of a few mm-size retrieved from the
corresponding cm-scale plugs were investigated by the digital rock visualization and simulations techniques.
**Table 1.** *Laboratory investigation methods and determined petrophysical characteristics*

| Experimental method | Determined petrophysical characteristics |
| --- | --- |



| 1.SEM | Mineral abundance, grain surface characterization of matrix and cementation |
|---|---|
| 2.Grain size analysis | Grain size distribution (*GSD*) |
| 3.X-ray diffraction (XRD) | Mineral components |
| 4.Gas porosimetry | Porosity ($\phi$) |
| 5.Gas permeametry | Permeability (1D) ($\kappa$) |
| 6.Mercury intrusion porosimetry (MIP) | Pore throat size distribution (*PTSD*), specific surface area (*SSA* - surface-to-bulk sample volume), characteristic length ($l_c$), pore throat length of maximal conductance ($l_{max}$), permeability ($\kappa$) |
| 7.Petrographic microscopy<br>Plane- (PPL) and cross- (XPL) parallelized and reflected- (RL) light microscopy, binocular (BINO). | Mineral abundance, grain surface characterization, cementation |
| 8.Extended computational workflow:<br><br>Image analysis | <br><br>Porosity ($\phi$), specific surface area (*SSA- surface-to-pore volume*), tortuosity ($\tau$), pore size distribution (*PSD*), connectivity index (*CI*), CT predicted porosity from MIP |
| Flow modelling | Permeability tensor ($\bar{\bar{\kappa}}$), tortuosity ($\tau$) |


**I. Petrographic** description of the rock composition and texture at the micro- scale:

- *Scanning Electron Microscopy (SEM, JCM-600, Bench Top Sem, Joel)* (Krinsley et al., 2005) and thin section optical microscopy (*Olympus BX53*) (Adams et al., 2017) were used to determine mineral abundance, grain surface characteristics of the matrix and cement.
- *Grain size distribution (GSD)* was determined by a Laser Diffraction Particle Size Analyzer (LS 13 320).

**II. X-ray diffraction** (XRD, *Miniflex 600, Rigaku*) was applied on powdered samples to determine their mineralogical composition.

**III. Petrophysical** laboratory measurements of effective rock properties

Effective porosity and permeability were evaluated on dried cylindrical samples (2.5 cm in diameter and 5-7 cm in length), following the RP40 procedure (see Practices for Core Analysis, 1998).



• *Effective Porosity* ($\phi$) was measured using a steady-state nitrogen gas porosimeter produced by
Vinci Technologies (*HEP-E, Vinchi Technology,* v3.20).
• *Permeability* (κ) was measured using a steady-state gas permeameter by Vinci Technologies
(Steady State Gas Permeameter for Educational Purpose: GPE 30, e.g. Tidwell and Wilson, 1997).
**IV.    Mercury intrusion porosimetry** (MIP, *Micromeritics AutoPore IV 9505*) was applied to dried
cylindrical samples of ~$1 cm^3$ to evaluate the following parameters (Table 1):
• *Pore throat size distribution (*Lenormand, 2003).
• *Specific surface area* (*SSA*, surface-to-bulk sample volume) (Rootare and Prenzlow, 1967; Giesche,

2006).

• *Characteristic length (*$l_c$)*:* the largest pore throat width (obtained from the increasing intrusion
pressure), where mercury forms a connected cluster (Katz and Thompson, 1987).
• *Pore throat length of maximal conductance (*$l_{max}$) (Katz and Thompson, 1987) defining a threshold
for pore throat size, $l$, where all connected paths composed of $l \geq l_{max}$ contribute significantly to
the hydraulic conductance, whereas those with $l < l_{max}$ may completely be ignored.
• *Permeability* ($\kappa$) (Katz and Thompson, 1987):
$$\kappa = \frac{1}{89} l_{max}^2 \frac{l_{max}}{l_c} \phi S(l_{max})$$                (1)
where $S(l_{max})$ is the fraction of connected pore space composed of pore throat widths of size $l_{max}$ and
larger.
**V.    Imaging and fluid flow modeling** of cylindrical samples of 1cm in diameter and 2 cm in length were
used, retrieved from the corresponding macroscopic plugs used in petrophysical lab measurements as
described above.
Extended computational workflow (the procedure is similar to that presented by Boek and Venturoli,
2010; Andrä et al., 2013) serves as the main methodology in our study (Fig. 2). It includes: 3D µ-CT imaging
of the porous samples; image processing and segmentation; statistical analyses for determination of
representative elementary volumes, and pore-scale flow modelling through the real 3D image of the rock.
-X-ray computed tomography (CT) (Fig. 2b)



The first step in this workflow, X-ray computed tomography (CT), produces a 3D image of a porous
rock. The resolution of μ-CT scanning was 2.5 µm cube voxel (isotropic), suitable for imaging those pore-
throats that effectively contribute to the flow in the studied type of sandstone (e.g. Nelson, 2009). Regions in
the raw 3D image having strong artefacts were removed so as to produce an image of 1180 voxels (2950µm)
edge size (Fig. 2b). The imaging was performed by using a *nanotom* 180S μ-CT device (GE Sensing &
Inspection Technologies, product line of Phoenix|x-Ray, Brunke et al., 2008) in the petrophysics laboratory at
the Leibniz Institute for Applied Geophysics (LIAG) in Hannover.
-Artefact removal and image segmentation (Fig. 2c)
Image artefacts produced by the CT scanning were reduced as described in Wildenschild (2013). The
beam hardening artefacts were removed by applying best-fit quadratic surface algorithm (Khan et al., 2016)
on each reconstructed 2D slice of the image. Ring artefacts reduction and image smoothing (with preservation
of sharp edge contrast) were performed using a non-local means filter (Schlüter, 2014).
Segmentation was performed in order to convert the grey-scale images obtained after the image filtering
to a binary image of volume pixels ("voxels"), to distinguish between the void and solid phases. The local
segmentation approach was used which considers the spatial dependence of the intensity for the determination
of a voxel phase, in addition to the histogram-based one (Iassonov et al., 2009; Schlüter et al., 2014). A two-
phase segmentation was performed by the converging active contours algorithm (Sheppard et al., 2004), a
combination of a watershed (Vincent et al., 1991), and active contour algorithms (Kass et al., 1988).

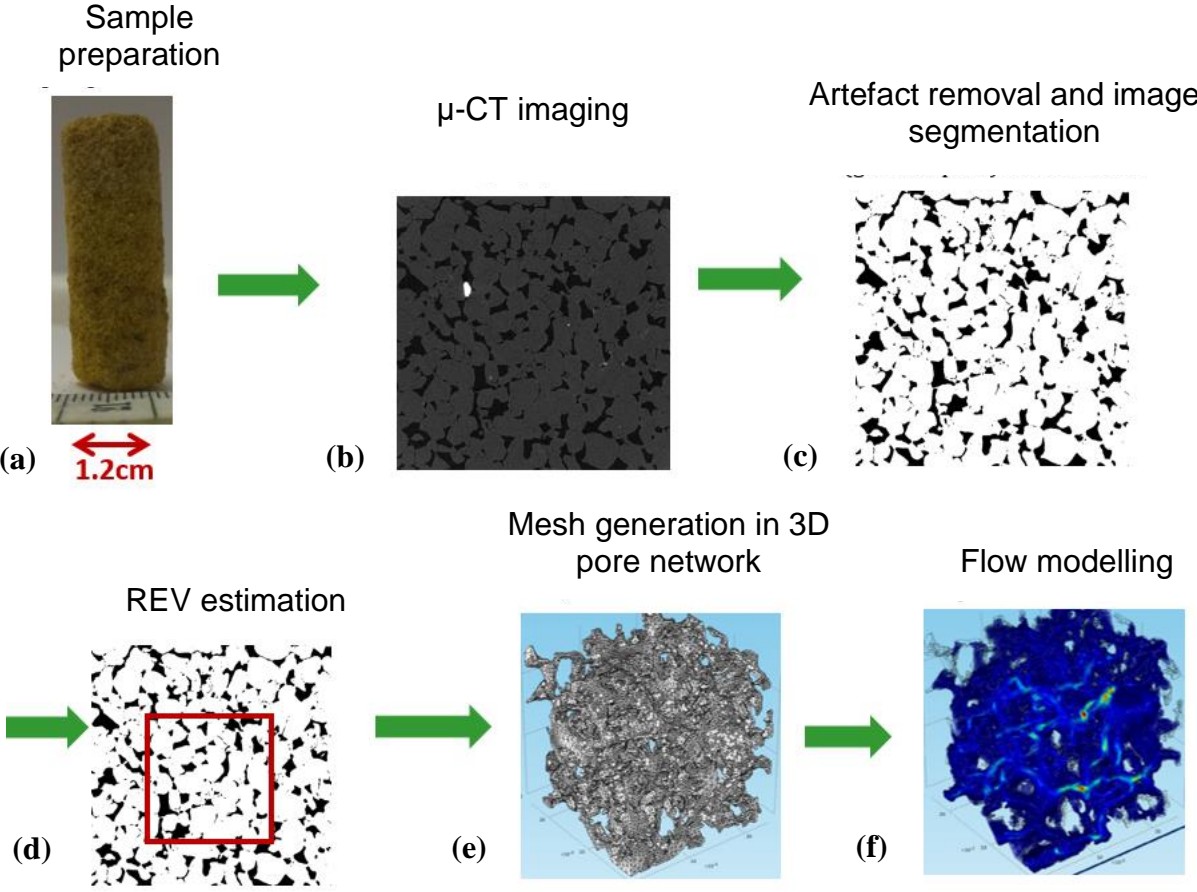

**Figure 2.** *Extended computational workflow. See text for more detail. Images (**e**) and (**f**) are adopted from Bogdanov et al. (2012).*

**-**Estimating a Representative Elementary Volume (REV) (Fig. 2d)

Simulations in the real-geometry of the imaged rock are computationally power- and time-consuming. Therefore, determination of REV (Bear, 1988) is required, assuming that at REV dimensions porous media are homogeneous. REV estimated for permeability would be required in the current study. However, multiple flow simulations (Blunt et al., 2013) at the pore-scale to upscale for permeability are computationally expensive. Instead, porosity, $\phi$, a basic macroscopic structural property of porous media, is usually used for the estimation of REV (Bear, 1988; Halisch, 2013; Tatomir et al., 2016), based on its correlation with permeability, $\kappa$, ($\kappa \sim \phi^3$; Kozeny, 1927; Carman, 1956).



Two approaches are used in our study to estimate REV (Halisch, 2013). In the "Classical" approach

REV for a scale is attained when porosity fluctuations in the sub-volumes growing *isotropically* in three
orthogonal directions become sufficiently small (Bear, 1988). Practically, a large number of randomly
distributed cubes were analysed through the entire 3D sample (1180 voxel edge length in our case) for their
image porosity (IP). Starting from a chosen cube size (10 pixel edge in our case), the cube size was increased
by 10-100 voxels. The REV size is specified when the agreement between the mean and median IP values, and
a saturation in IP fluctuations, are attained.

Alternatively, the "Directional" REV approach can capture porosity changes in a specific direction,

which are caused by microscopic structural features, such as grain packing, cracks, texture effects, etc.
(Halisch, 2013). Average porosity is first calculated slice-by-slice across the segmented image in each (x-, y-
and z-) direction. Then, variogram analysis (Cressie, 1993) is used to describe a degree of spatial variability in
porosity in each direction, based on the assumption that a distance at which no spatial correlation exists reflects
a scale of homogeneity, which defines the REV. The variogram $\hat{\gamma}(h)$, i.e. the expected squared difference
between two observations (averaged slices porosity), is calculated as a function of their separation distance, *h*
*(lag)*. Practically, the lag distance where the variogram saturates is that distance at which no spatial correlation
exists (the range). Depending on the sample heterogeneity across the scales, the variogram may manifest the
range for each scale.

**-**Mesh generation in 3D pore network and flow simulation (Figs. 2e-f)

The binary 3D REV (regular grid, raster-) image of the pore space is spatially discretised by tetrahedrals

with Materialize software (Belgium). This step is required for the import to the FEM-based modeling software,
Fig.2e. Stokes *(creeping) Flow (Re << 1)* is simulated in the pore network (Fig. 2f) by the following equations
(e.g. Narsilio et al., 2009; Bogdanov et al., 2011):

Stokes equation: $-\nabla p + \mu \nabla^2 \bar{u} = 0$                                           (2)

Continuity equation:        $\nabla \cdot \bar{u} = 0$                                               (3)

where $\nabla p$ is the local pressure gradient, $\bar{u}$ is local velocity field in the pore network, μ is fluid dynamic

viscosity.




Fixed pressures, *p=const,* were specified at the inlet and outlet boundaries of the fluid domain. At the
internal pore walls and at the lateral domain boundaries, no-slip boundary conditions are imposed ($\bar{u} = 0$) (e.g.
Guibert et al., 2014). These also simulate the experimental flow setup in a steady-state permeameter (e.g.
Renard et al., 2001). FEM based Comsol Multiphysics simulation environment, v5.2a, is utilized.
• *Upscaling to macroscopic permeability tensor,* $\bar{\bar{\kappa}}$
Macroscopic velocity $< \bar{v} >$ is evaluated by volumetric averaging of the local microscopic velocity
field (e.g. Narsilio, 2009; Guibert et al., 2014). Then, from three average macroscopic velocity vectors ($v_{ij}$),
corresponding to the imposed pressure gradients in x-, y- and z- directions, the full second-rank upscaled
permeability tensor, $\bar{\bar{\kappa}}$, in 3D is derived:
$$< \bar{v} >= -\frac{1}{\mu\phi} \bar{\bar{\kappa}} \bar{\nabla} p \qquad (4)$$
by solving the following linear system of equations for $\bar{\bar{\kappa}}$:
$$\begin{pmatrix} v_{xx} & v_{xy} & v_{xz} \\ v_{yx} & v_{yy} & v_{yz} \\ v_{zx} & v_{zy} & v_{zz} \end{pmatrix} = -\frac{1}{\mu\phi} \begin{pmatrix} \kappa_{xx} & \kappa_{xy} & \kappa_{xz} \\ \kappa_{yx} & \kappa_{yy} & \kappa_{yz} \\ \kappa_{zx} & \kappa_{zy} & \kappa_{zz} \end{pmatrix} \begin{pmatrix} \nabla p_x & 0 & 0 \\ 0 & \nabla p_y & 0 \\ 0 & 0 & \nabla p_z \end{pmatrix} \qquad (5)$$
Permeability tensor is symmetrized by:
$$\bar{\bar{\kappa}}_{sym} = \frac{1}{2}(\bar{\bar{\kappa}} + \bar{\bar{\kappa}}^T) \qquad (6)$$
• *Tortuosity* ($\tau$), is calculated separately in x-, y- and z- directions in the meshed domain using a

particle tracing tool of Comsol Multiphysics software, after averaging the multiple paths.

-3D image analysis is conducted on a high quality full segmented μ-CT image (of 1180 voxel (i.e. 2950
μm) size). Non-connected void clusters of the specimen are labelled, then separation of the cluster into objects
is performed using the distance map watershed algorithm (e.g. Brabant et al., 2011; Dullien, 1992). Image
analysis operations were assisted by Fiji-ImageJ software and plugins (Schindelin et al., 2012).
The following geometrical descriptors are derived:
• *CT predicted porosity* is evaluated on the segmented image by ImageJ software (Table 1).



• _Pore specific surface area of the segmented image (SSA - surface-to-pore volume)_ is evaluated

using ImageJ software, when pore volume is calculated for pores larger than resolution limit of 2.5

302       µm.

• _Tortuosity ($\tau$)_ (Bear, 1972; Boudreau, 1996) is evaluated in x-, y- and z- directions on 3D

segmented image by finding the average of multiple shortest paths through the main pore network

using the Fast Marching Method (Sethian, 1996).

• _Pore size distribution (PSD)_ is specified, when pore size is described by Ferret maximum calliper

(e.g. Schmitt et al., 2016).

• _Connectivity Index_ (_CI_): Euler characteristic ($\chi$) is a topological invariant (Wildenschild and

Sheppard, 2013; Vogel, 2002). Because the number of pore connections depends on the number of

grains, to compare connectivity between the three samples which have the same specimen sizes but

different grain sizes, we suggest using a _Connectivity Index_, computed by dividing Euler

characteristic by a number of grains in the specimen, _N_ (after Scholz et al., 2012).

$$CI = \frac{\chi}{N} \tag{7}$$
• _CT predicted porosity at the image resolution size from MIP_: We propose a new simple method to

estimate the image porosity at a given resolution. Multiplying the mercury effective saturation at

the µ-CT resolution (e.g. Fig.7a, red dashed line) by porosity of the same sample measured by gas

porosimeter, yields µ-CT-predicted image porosity at resolution limit.

**3.    Results**

**3.1. Petrographic and petrophysical rock characteristics**

In this section all three types of sandstone rocks are characterised by the techniques 1-8 listed in Table

1. The results are presented in Figs. 3-10 and summarised in Tables 2 and 3.

**Sample S1**: The top unit layer of ~1.5 m thickness (Fig. 1c) consists of yellow-brown sandstone (Fig.

3a), moderately consolidated. The sandstone is a mature quartz arenite (following Pettijohn et al., 1987) with
minor Fe-ox, feldspar and heavy minerals (Fig. 3b). The grain size distribution has a mean grain size of 325
µm (Fig.6a). The grains are moderately sorted (according to classification of Folk and Ward, 1957) (Table 2),





sub-rounded to well-rounded, with local mm-scale thick darker envelopes (Fig. 3a, b). The sandstone consists
of mm-scale alternating layers of large and small sand grains. Secondary silt (~ 45 µm) and clay (~0.95 µm)
populations are detected in grain size distribution (Fig.6). X-ray diffraction detected small amount of kaolinite.
The Fe-ox grain-coating and meniscus-bridging cement is composed of overgrown flakes aggregated into ~10
µm size structures (Fig. 3c-3e). Mn-ox is evident too but is rear (Fig. 3d).

The pore network is dominated by primary inter-granular well interconnected macro- porosity (Fig. 3b).

However, sealed and unsealed cracks in grains are also observed. Higher Fe-ox cementation at mm- scale on
horizontal planes is recognized (also shown in Fig. 3a). In addition, smaller voids between Fe-ox aggregates
and flakes are found at a µm scale and smaller (Fig.3 c-e).

From the pore throat size analysis conducted with MIP, 82% of pore volume are macro-pores (>10 µm),

with log-normal distribution with a peak at 44µm (Fig.7). The characteristic length, i.e. the largest pore throat
length where mercury forms a connected cluster is $l_c = 42.9$ µm (Fig.8), and pore throat length of maximal
conductance is $l_{max} = 34.7$ µm (Fig. 9). Porosity evaluated by laboratory gas porosimetry varies in the range
of 26-29% for 7 different samples of S1 (Fig. 10). Multiplying the mercury effective saturation (85.8%) at the
µ-CT resolution (2.5 µm) (Fig. 7a, red dashed line) by porosity of the same sample measured by gas
porosimetry (27.36 %), yields µ-CT predicted image porosity of 23.5 % at resolution limit of 2.5 µm (Table

2).

Permeability evaluated by laboratory gas permeameter has an average of 350 mD (range of 130-500

mD) for 5 samples measured perpendicular to the depositional plane (z-direction), and 640 mD for 2 samples
measured parallel to depositional plane (x-y directions) (Fig. 10). Permeability from the MIP measurement
(Katz and Thompson, 1987) (see Sec.2.2) reached 330 mD (Table 2).









**Figure 3.** *Sample S1. (a) A plug analysed by petrophysical methods, and from which thin sections were*

*extracted. Darker laminae in x-y plane at millimetre scale are observed. (b) Thin section: Quartz grains (pink)*

*show interlocking and interpenetration textures indicative of compaction and pressure solution. Grain size*

*variations reveal laminae of larger grains, deposited on the top of laminae of smaller grains. Empty pores are*



*in yellow. (c) Scanning Electron Micrograph: Grain-coating and meniscus-bridging cement and overgrowth*
*of Fe-ox flakes. (d-e) Thin section zoom-in view of (c): at 5 μm and 2 μm scale, respectively. (f-h) At the same*
*field of view in PPL, XPL and RL, respectively (see Table 1 for specification). (i) Fe-ox flakes (yellow) on*
*quartz grains (pale grey).*

**Table 2.** *Petrophysical characteristics of the three studied sandstone layers*

| | Method | S1 | S2 | S3 |
|---|---|---|---|---|
| **Grain size** | Laser diffraction | 325 μm<br>**medium Sand**<br>**moderately sorted**<br>sand:   92.6%<br>silt:    6.6%<br>clay:   0.8% | 154 μm<br>**very fine sand**<br>**poorly sorted**<br>65.7%<br>31.3%<br>3% | 269 μm<br>**fine Sand**<br>**moderately sorted**<br>94.4%<br>4.8%<br>0.8% |
| **Pore throat size** | MIP | Mode 1: 44 μm<br>Mode 2: 0.035 μm<br>Mode 3: 2.2 μm<br><br>**macro pores**<br>**well sorted** | 0.035 μm<br>3.5 μm<br><br>**sub-micro**<br>**pores**<br>**poorly sorted** | 35 μm<br>0.035 μm<br>2.2 μm<br><br>**macro pores**<br>**well sorted** |
| **Pore size** | Image analysis<br>(min. object size<br>2.5 μm) | 194 μm<br>(*FWHM<br>[150,335] μm) | Mode 1: 21 μm<br>Mode 2: ~100<br>μm | 223 μm<br>(*FWHM<br>[145,400] μm) |
| **Characteristic length, $L_c$** | MIP | 42.9 μm | 12.3 μm | 36.9 μm |
| **$l_{max}$ contributing to maximal conductance** | MIP | 34.7 μm | 8 μm | 31.4 μm |
| **Porosity, φ** | gas | 28 ± 2 % | 19 ± 5 % | 31 ± 1 % |
| | CT predicted image porosity from MIP | 23.5 % | 6.65 % | 30.4 % |
| | **μ-CT segmented** | 17.52% | 6.89% | 28.32% |
| **Permeability, κ [mD]**<br>⊥ - perpendicular to    layering    (z- | gas | ⊥ 350<br>∥ 640 | ⊥ 2.77<br>∥ 7.73 | ⊥ 220<br>∥ 4600* |
| | MIP | 330 | 4 | 466 |



| direction) ‖ - parallel to layering (x-y plane) | Flow modelling | $\begin{pmatrix} 420 & 66.3 & 1.91 \\ 66.3 & 344 & 12.8 \\ 1.91 & 12.8 & 163 \end{pmatrix}$ | - | $\begin{pmatrix} 4517 & 5 & 38 \\ 5 & 4808 & 547 \\ 38 & 547 & 4085 \end{pmatrix}$ |
|---|---|---|---|---|
| **Specific surface area, SSA** | MIP (surface-to-bulk-volume) | $3.2\ \mu m^{-1}$ | $12.2\ \mu m^{-1}$ | $0.16\ \mu m^{-1}$ |
| | μ-CT at 2.5 μm resolution size (surface-to-pore-volume) | $0.068\ \mu m^{-1}$ | $0.136\ \mu m^{-1}$ | $0.069\ \mu m^{-1}$ |
| **Connectivity index** | Image analysis | 3.49 | 0.94 | 10 |
| **Tortuosity, τ** | Flow modelling | - | - | x: 1.443 y: 1.393 z: 1.468 |
| | μ-CT shortest path analysis | x: 1.385 y: 1.373 z: 1.477 | - | x: 1.316 y: 1.338 z: 1.394 |


Legend:
gas – gas porosimetry/permeametry
MIP - mercury intrusion porosimetry
FWHM - full width at half maximum, log-normal distribution.
*Addressed in the Discussion Sect.

**Sample S2**: The intermediate unit layer of ~20 cm thickness consists of grey-green moderately
consolidated sandstone (Figs. 1c, 4), composed of sub-rounded to rounded very fine sand grains (154 μm),
and poorly sorted with 35 % of the particles of silt and clay (Fig. 6, Table 2). Secondary silt (~ 40 μm), sand
(400 μm) and clay (1.5 μm) populations are also detected. The grains are composed of quartz with minor Fe-
ox coating the grains and also minor quantities of heavy minerals (Fig. 4d). Clay filling the pore space was
identified by XRD as a kaolinite mineral. It appears as a matrix, being grain-coating, meniscus-bridging, and
pore-filling (Fig. 4b, c). Therefore, the unit layer (Fig. 1c) is classified as quartz wacke sandstone.
The pore network is influenced by the extent of clay deposition on coarser grains, identified mostly in
laminae (Fig. 4a, d). Yet, inter-granular connectivity of macro pores can still be recognized (Fig. 4b, c). The





effective pore network consists of inter-granular macro-pores distributed between the laminae or zones richer
in clay and Fe-ox.

Integration of results of grain size and pore throat size analyses (Figs. 6, 7) confirms that the reduction

of inter-granular pore space in S2 is due to clay matrix, which is reflected in the poor grain sorting and large
variance in pore size. In the pore throat size analysis (Fig.7) only 15 % of pore volume is in macro pores that
are larger than 10 µm. The prominent sub-micron pore mode is of ~35 nm, with population containing ~45 %
of the pore volume. This population of pores occurs inside the clay matrix. The secondary population of pore
volume is poorly distributed within the range of 0.8-30 µm. The peak at 350 µm (Fig. 7b) is probably due to
disintegration of the sample during preparation. Characteristic length (Sect.2.2), $l_c = 12.3$ µm (Fig. 8), and
pore throat length of maximal conductance, $l_{max} = 8$ µm (Fig. 9) (both are with a large error resulting from
the uncertainty in threshold pressure), suggest a connectivity of macro pores regardless of their small fraction
in the total pore space. Porosity of S2 evaluated for 8 different samples varied in the range of 14.5-23.5 %
(Fig.10). From PTSD (Table 1) and gas porosimetry (for a sample of 18.6% porosity), µ-CT predicted an
image porosity at resolution limit of 2.5 µm of 6.65 % (Table 2). Gas permeability measured in z-direction
was calculated for 5 samples (Fig.10): in four of them permeability ranged within 1-12 mD, increasing with
porosity. However, one sample was with an exceptionally large porosity and permeability, 23 % and 62 mD,
respectively. Permeability measured for 3 samples in x-y plane ranged within 4-12 mD, showing also ~15 %
of porosity (Fig. 10). In addition, for the samples with ~15 % porosity, permeability was ten-fold larger in x-
y plane (parallel to the layering) than in z-direction (perpendicular to the layering). Permeability derived from
the MIP reached 4 mD, which agrees with an average of 2.77 mD and 7.73 mD (Table 2) measured in z-
direction by gas permeameter (excluding one exceptionally high value, Fig. 10).








**Figure 4.** *Sample S2. (a) A plug analysed by petrophysical methods, and from which thin sections were*

extracted. Prominent features are dark and yellowish zones. **(b)** *The dark laminae is richer in clays and iron*

oxides that seal and occlude intergranular space in a specific horizon, whilst above and below the macro

pores are mostly empty. This pattern is repetitive on mm and cm scales (Fig.4a). **(c)** *Clay and silt accumulated*



*as meniscus (M), and as clay matrix (CM). P refers to open pores. (d) Pore clogged by clay and iron oxide.*
*(e) Rock texture under binocular. Clay matrix is in white, quartz grains are in pale grey.*
**Sample S3**: Samples were taken from the ~1.5 m thick bottom layer in the outcrop (Fig. 1c) consisting
of (pale) red-purple poorly consolidated sandstone with grains covered by secondary red patina (Fig. 5). It is
composed of friable to semi-consolidated fine (269 µm) moderately sorted sand (Table 2), where only 5.6 %
of particles are silt and clay (Fig. 6). Secondary silt (~ 50 µm) and clay (~ 0.96 µm) populations were also
detected. The sandstone consists of sub-rounded to rounded grains showing a laminated sedimentary texture,
of cyclic alternation of darker and lighter red bands of millimetre scale thickness (Fig. 5a). The dark laminae
contain slightly more cementation of Fe-ox meniscus and pore filling cement (Fig. 5b). This bed consists of
ferruginous quartz arenite. The grains are dominated by quartz with very minor feldspar and black opaque
mineral grains perhaps Fe-ox. X-ray diffraction indicated $SiO_2$ mineral only. The Fe-ox coating of grains is
less extensive than in other samples. Pore interconnectivity in this sandstone is high (Fig. 5b, c). Heavier
cementation is rarely observed (Fig. 5c), organized in horizontal laminae. Features including grain cracks,
grain to grain interpenetration, and pressure solution are recognized too (Fig. 5d). Pore throat size analysis
showed that 95 % of the pore volume is presented by macro-pores (Fig. 7), which agrees with the minority of
fine particles. Characteristic length and pore throat length of maximal conductance are $l_c = 36.9$ µm and
$l_{max} = 31.4$ µm (Figs. 8-9).
Porosity measured by laboratory gas porosimeter varies in the range of 30-32% for 4 different samples
(Fig.10). From PTSD and gas porosimetry (Figs. 7 and 10), µ-CT predicted image porosity at resolution limit
of 2.5 µm is 30.4 % (Table 2). Permeability measured by laboratory gas permeameter yields an average of
220 mD for 2 samples measured in z-direction, and 4600 mD for 2 samples measured in the x-y plane,
showing a ten-fold difference (analysed in Discussion Sect.). Permeability derived from the MIP reached 466
mD (Table 2).






**Figure 5.** *Sample S3. (a) Plugs analysed by petrophysical methods, from which thin sections were extracted.*

*Laminae are recognized by their slightly dark and red colour. (b) General view under a binocular microscope*





*reveals red laminae ~500 μm thick. (**c**) A millimetre-scale lamina is indicated by enhanced of Fe-ox*

*cementation of meniscus-type and partly by inter-granular fill. Grain surfaces are coated by thin Fe-ox. Black*

*and orange cements represent crystalized and non-crystallized Fe-ox, correspondingly. Some cracked grains*

*are observed, sporadically cemented by Fe-ox. (**d**) Partialy dissolved grains are coated by cement. (**e**) High*

*resolution observation of a clear grain by binocular.*

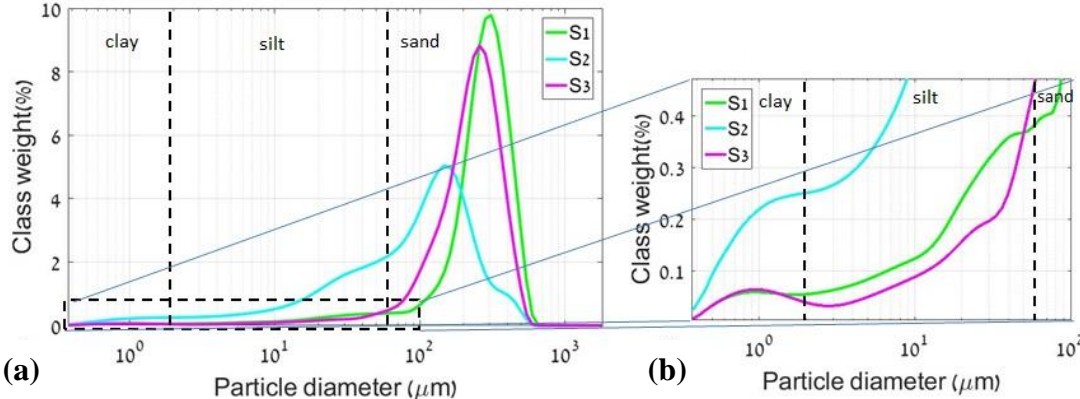

***Figure 6.*** *(**a**) Grain size distribution. (**b**) Zoom-in into grain size distribution in the fine grain size region*

*plotted for samples S1 (green), S2 (blue) and S3 (purple). Samples S1 and S3 have unimodal distribution*

*(main mode sizes are at 325 μm and 269 μm, respectively), being moderately sorted with small skewness tail.*

*Sample S2 (main mode size is at 154 μm) has a multi-modal distribution, being poorly sorted. Classification*

*is by Folk and Ward (1957).*

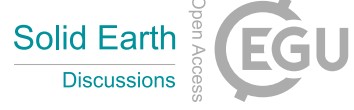



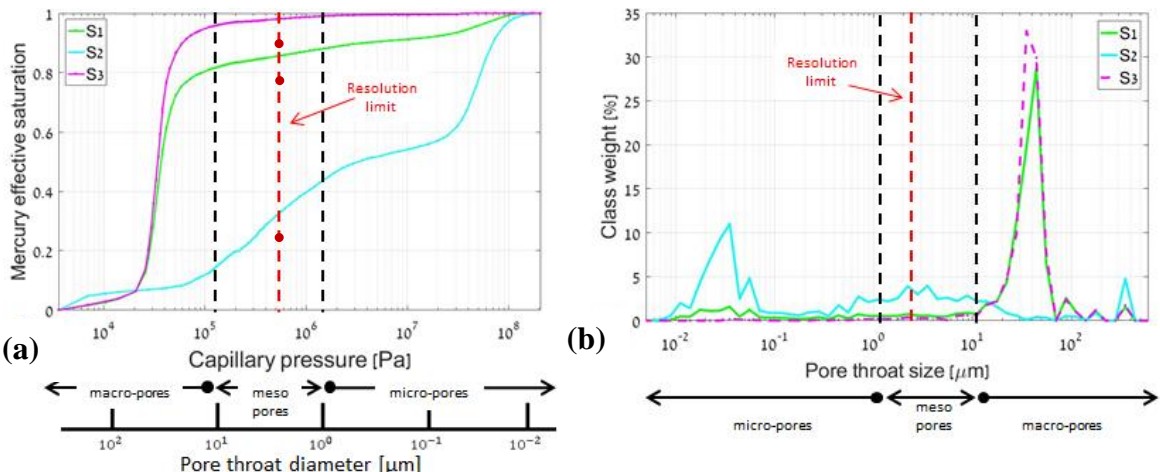

**Figure 7.** *Pore throat size cumulative (**a**) and pore throat distribution (**b**) of the samples. Samples S1 and S3 have unimodal distribution (main mode sizes are at 44 µm and 35 µm, respectively). Sample S2 has a multimodal poorly sorted distribution: a wide population is distributed within a range of 0.8-30 µm, and another population within a range of 0.008-0.08 µm. Pore throat sizes larger than 100 µm in MIP may result from disaggregation of grains during sample preparation. Black dashed lines separate the region to macro- (>10 µm), meso- (1-10 µm) and micro- pore throats (<1 µm). Resolution limit of the µ-CT imaging is presented by the red dashed line to indicate the fraction of the pore space that could be resolved. The horizontal axis scale is log-normal.*



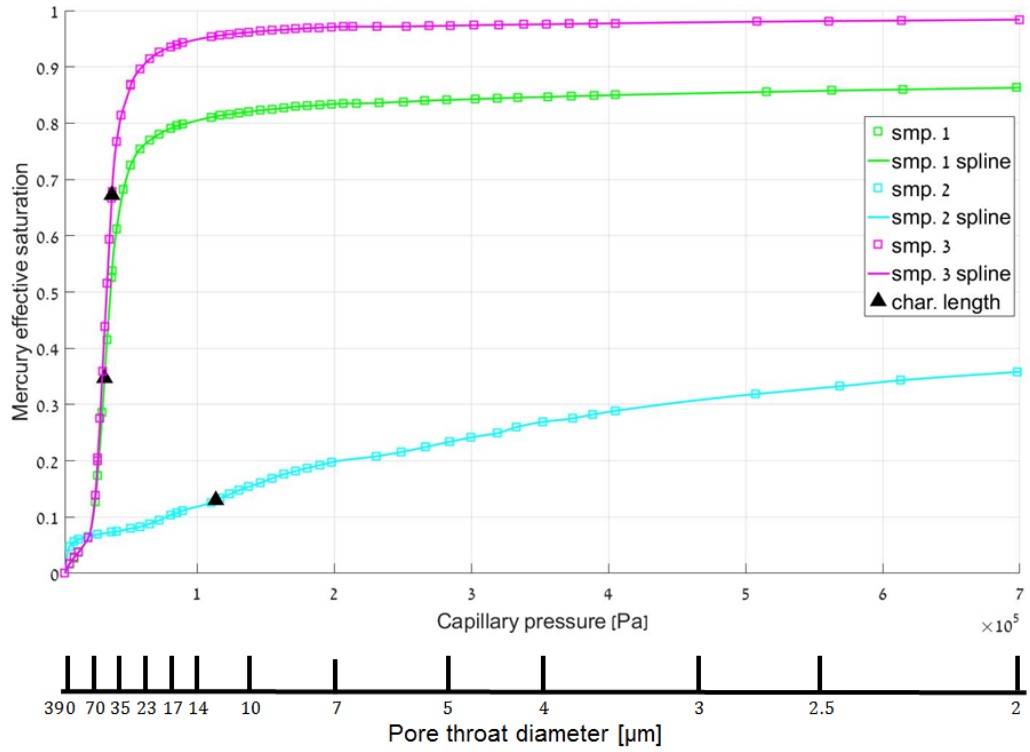

**Figure 8.** *Mercury saturation vs. capillary pressure in the mercury intrusion measurements is plotted. A spline curve was used to fit the data. The triangles assign the pressure corresponding to the maximum slope of each curve, a threshold pressure, at which mercury first forms a connected path spanning the sample (Katz and Thompson, 1987). The threshold pressure, in turn, corresponds to pore throat size termed a characteristic length, $l_c$ (see Sect.2.2).*





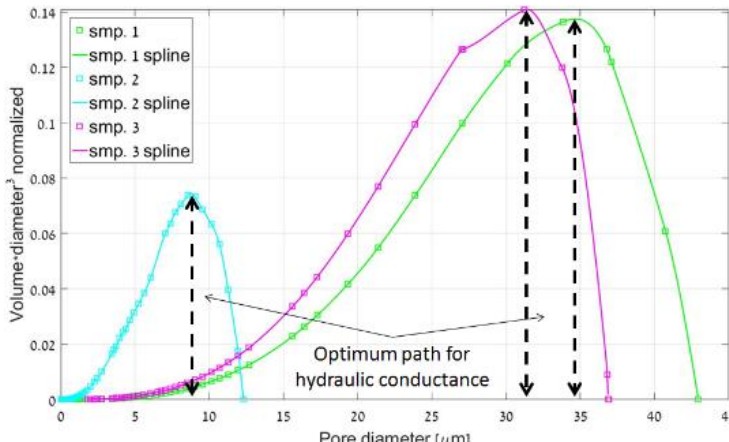

*Figure 9.* *The pore throat length of maximal hydraulic conductance, $l_{max}$, is defined from the maximal*

*contribution to (normalized) hydraulic conductance (Katz and Thompson, 1987), specified at the vertical axis*

*of the chart. The corresponding pore throat diameter (at x-axis) specified by black arrow assigns pore throat*

*diameter (or pore throat length of maximal conductance), $l_{max}$, where all connected paths composed of $l \geq$*

*$l_{max}$ contribute significantly to the hydraulic conductance (see Sect.2.2).*




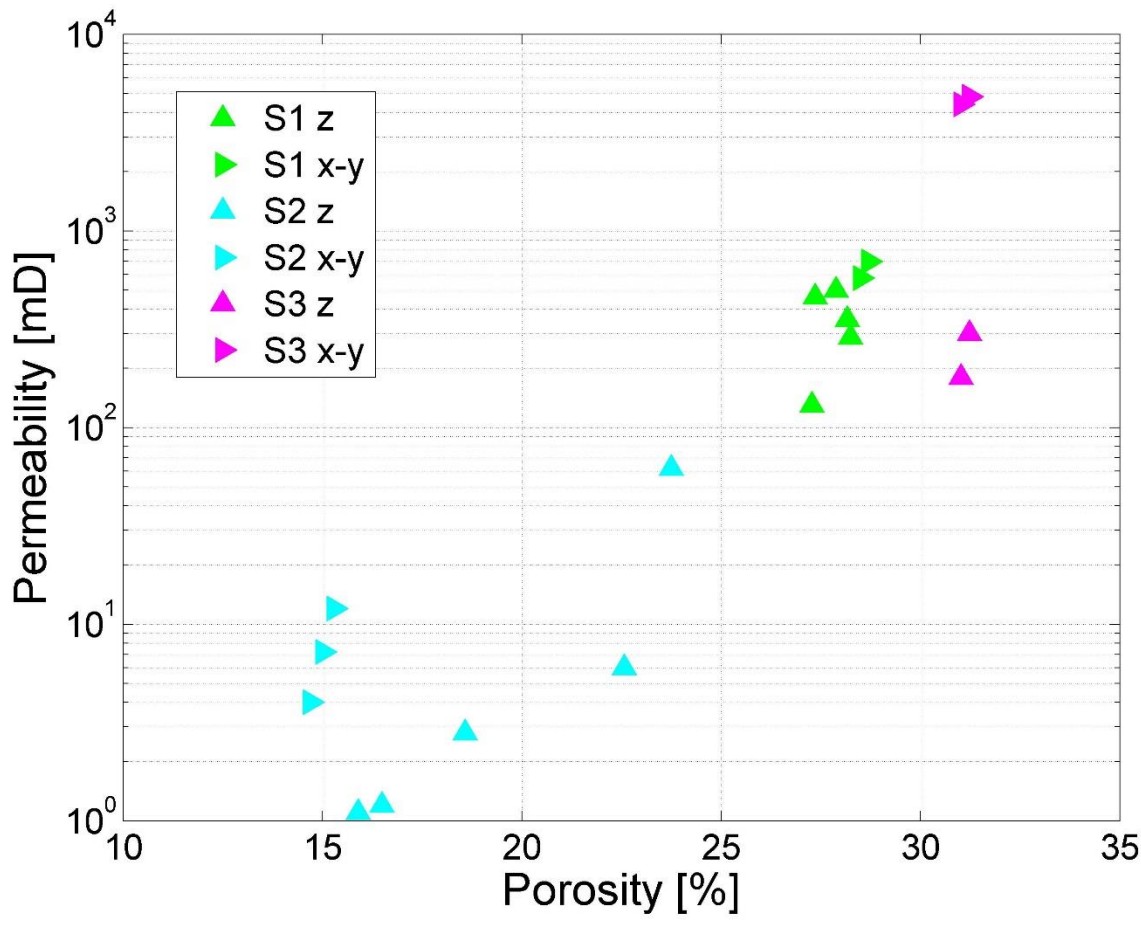


***Figure 10.*** *Results of porosity-permeability lab measurements. Permeability of the samples was measured in*

*directions perpendicular to the bedding (z-direction) and parallel to the bedding (x-y plane) and is presented*

*in the log-scale. Correlations between porosity and permeability is observed.*


**3.2. Rock characterization with extended computational workflow**

The plugs from the three samples, which were analysed in the lab for porosity and permeability (Table

2), were scanned at 2.5 μm resolution by μ-CT scanner (Fig. 2b). Then, image processing and segmentation



(Sect. 2.2) were performed to produce a binary (grains-pore system) 3D image (Fig. 2c). This step was
followed by the determination of representative elementary volume (REV, Fig. 2d) by classical and directional
approaches, estimated here for the porosity (see Sect. 2.2 for more detail).
**Classical REV** (presented by Figs. A1-A3 in the Appendix A).
For Sample S1 homogeneity was attained at 475 voxels (1187 µm) sub-volume size, when the difference
between the median and mean porosities dropped below 0.1 % (Fig. A1 in the Appendix A). Resulting average
image porosity (IP=17.52 %) is lower than the lab porosity, 27.36 %, measured on the same macroscopic
sample (Table 2). This is expected, as pores smaller than resolution limit of 2.5 µm of µ-CT image are assigned
as grain voxels. The µ-CT predicted porosity at resolution limit (derived with MIP) for S1 is 23.5%, which is
still 6 % higher than the image porosity (Table 2).
For Sample S2 the mean and median converged at 950 voxel (2375 µm) sub-volume size only (Fig. A2
in the Appendix A), which approaches the size of the entire sample (1180 voxels (2950 µm)), although the
scattering remained high (6.3 % and 7.8 % for min and max porosity, respectively). As a result, it is suggested
that homogeneity cannot be attained by the classical REV approach for S2 as a whole. IP is only 6.89%
compared to 18.6 % average porosity measured on the same sample in the lab (Fig.10, Table 2). The IP is
close to the µ-CT predicted porosity of 6.65 % at resolution limit of 2.5 µm (derived with MIP).
For Sample S3 the mean and median converged at 350 voxel (875 µm) sub-volume size (Fig. A3 in the
Appendix A), where the scattering dropped to 4 %, and a homogeneity is suggested. IP of 28.32 % is close to
of 30.4 % predicted by MIP at resolution limit of 2.5 µm, and to 31.5 % measured in the lab at the same
macroscopic sample (Table 2).
**Directional REV** (Figs. 11-14).
Sample S1 (Figs. 11 and 12): Slice-by-slice porosity analysed in three directions in every segmented
sample scanned with resolution 2.5 µm, distinguishes the z-direction as having an exceptional behaviour (Fig.
11a-c). Specifically, the difference between the maximum and minimum porosity is 7.44 % in the z-direction,
in contrast to 5.7 % in the x- and y- directions, which agrees with std. of 1.83 %, relative to 1.24 % and 1.01
% in x- and y-directions, respectively. Moreover, in z-direction median is higher than the mean porosity, in
contrast to that in x- and y- directions. It is seen that slice-by-slice porosity in x- and y- directions shows





fluctuations around the representative mean values (Fig. 11 a, b), due to changes in grains cross section
position. However, this behaviour is perturbed in the z-direction (Fig. 11c) close to slice #250, where two
domains with different IPs are recognized in both sides of that slice. A sub-domain of 0-250 (575 μm) slices
have ~15% of mean porosity, in contrast to sub-domain of 250-1180 slices of ~18 % of mean porosity (the
median is higher than the mean because of the higher number of the slices in the range 250-1180).
From the variogram analysis the representativeness is reached at ~100 voxel edge length (250 μm) in
the x- and y- directions, respectively, where in the z-direction it is reached at 350 voxel edge length (875 μm)
(Fig. 11d-f). Alternatively, the REV determined using the classical approach, was 475 voxel edge length (1187
μm).
In addition, the plug of the Sample 1 was scanned also at 5 μm resolution (in addition to 2.5 μm
resolution presented above, Fig.11) that allows investigation of a specimen of size 7145 x 7145 x 9330 μm
(Fig. 12). This image resolution is also appropriate based on porosity and pore network connectivity because
the volumes of the injected mercury are very similar at pore throat size of 2.5 μm and 5 μm (Fig.9) as well as
the mercury effective saturations (Fig.8). Fig. 12 shows an additional scale of porosity fluctuations for the
larger sample (i.e. 7145 x 7145 x 9330 μm). The range ~2000 μm in z-direction, associated with a half of
cycle of porosity fluctuations, indicates that both high and low porosity bands appear in the considered volume,
separated by this distance. Therefore, based on the larger range observed (Fig.12f), the whole 2950 μm edge
size cube of S1 scanned with 2.5 μm resolution is chosen for the flow modelling.



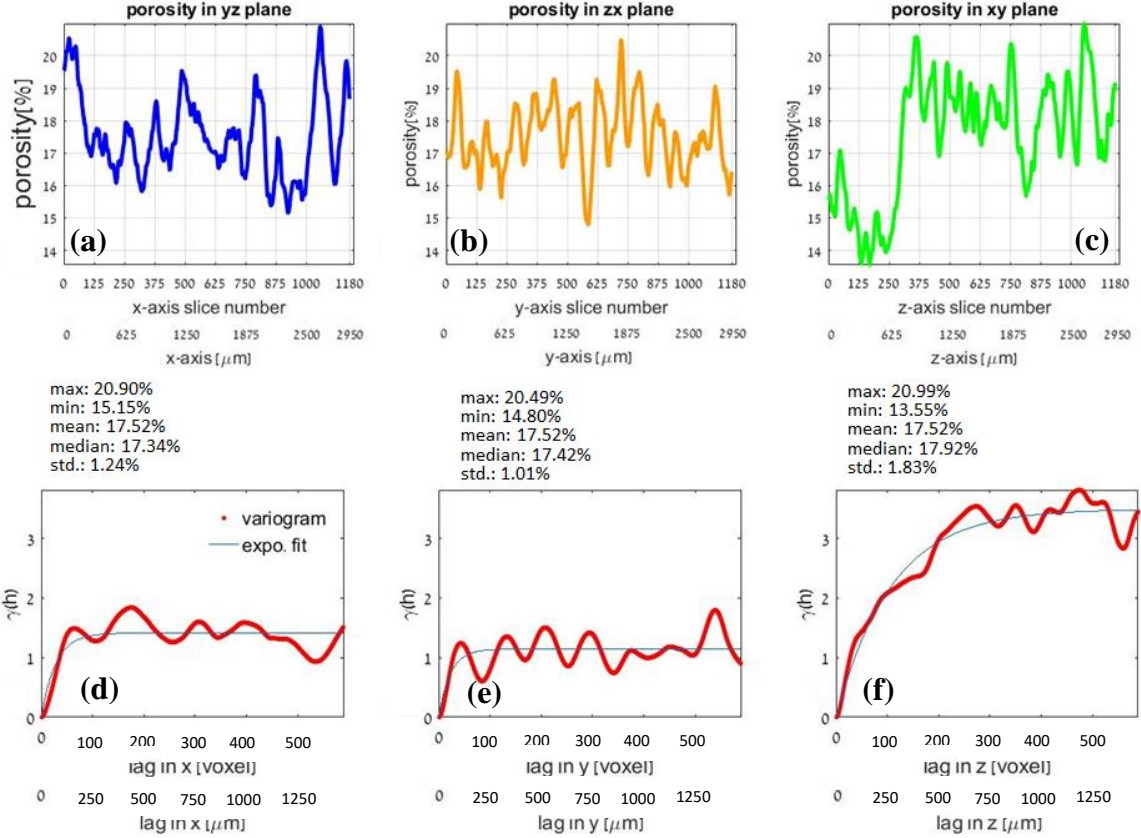


**Figure 11.** *Directional REV analysis for Sample S1 scanned at 2.5 μm resolution. In the top row porosity calculated slice-by-slice for the x-, y- and z- directions is presented (**a-c**). At the bottom row (**d-f**) the conducted variogram analysis indicates representativeness reached at 100 voxel (250 μm) edge length in x- and y- directions, respectively, where in z-direction at 350 voxel (875 μm), shown by the range of the variogram saturation values. The cyclicity in the variogram refers to cyclicity of the porosity at the pore scale.*

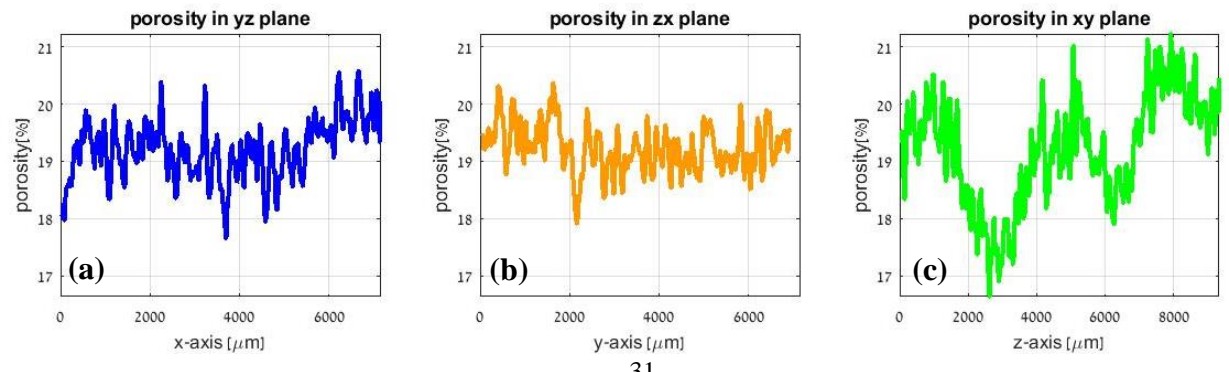

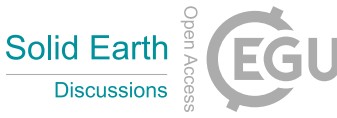

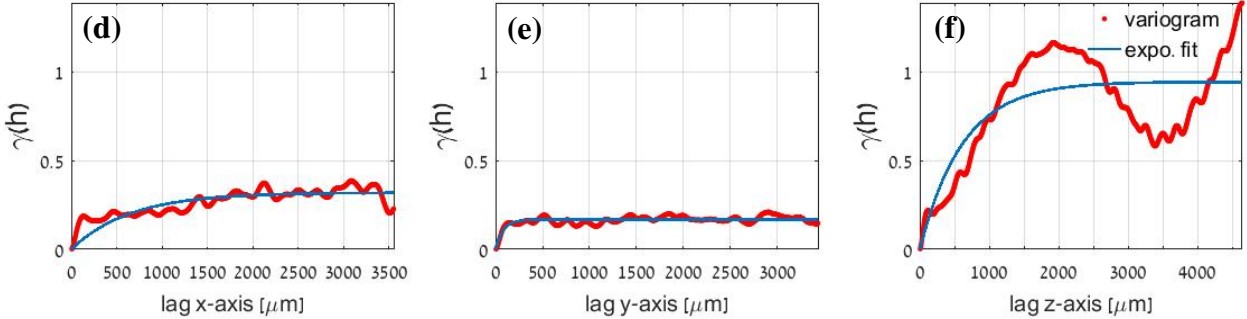

*Figure 12. Directional REV analysis for Sample S1 scanned with 5 μm resolution, on the domain larger than that studied in Fig. 11 (see text for explanations). In the top row porosity calculated slice-by-slice for the x-, y- and z- directions is presented (a-c). At the bottom row (d-f) conducted variogram analysis shows cyclicity in the x- and y-directions associated with the porosity fluctuations at the pore scale. In the z-direction the range ~2000 μm is associated with porosity fluctuations between the high and low porosity bands separated by this distance.*

Sample S2 (Fig. 13): Although the median for all the directions approaches the mean, still each direction
shows a remarkably different trend (Fig. 13a-c). The largest difference between minimum and maximum slice
porosity, 6.57 %, appears in the z- direction, compared to 4.5 % and 3.56 % for x- and y- directions,
respectively. The standard deviation in the z-direction (1.53 %) is about double than in other directions (0.86
% and 0.73 %). An increase in porosity in z-direction is observed, accompanied also by a reduction of
fluctuations around the mean, as it would be expected from a homogeneous porous media. This increasing
porosity trend in z-direction is in inverse correlation with the content of clay matrix between the sand grains
(Fig. 13a-c, brown curve; Fig. 4b, c). This anisotropic effect is prominent in z-direction.
From the variogram analysis, the representativeness in x-direction is reached for the large cube edge
size of 500 voxel (1250 μm), but for the y- direction it is not reached at all (Fig. 13d-f). However, the most
uncorrelated distribution of pores is in the z- direction, where saturation is not reached too, and a fit is still
presented by an inclined straight line. Therefore, based on the above analysis, REV could not be achieved
within the CT-scanned sample S2, which also agrees with result of the classical REV analysis (Fig. A2 in the
Appendix A).





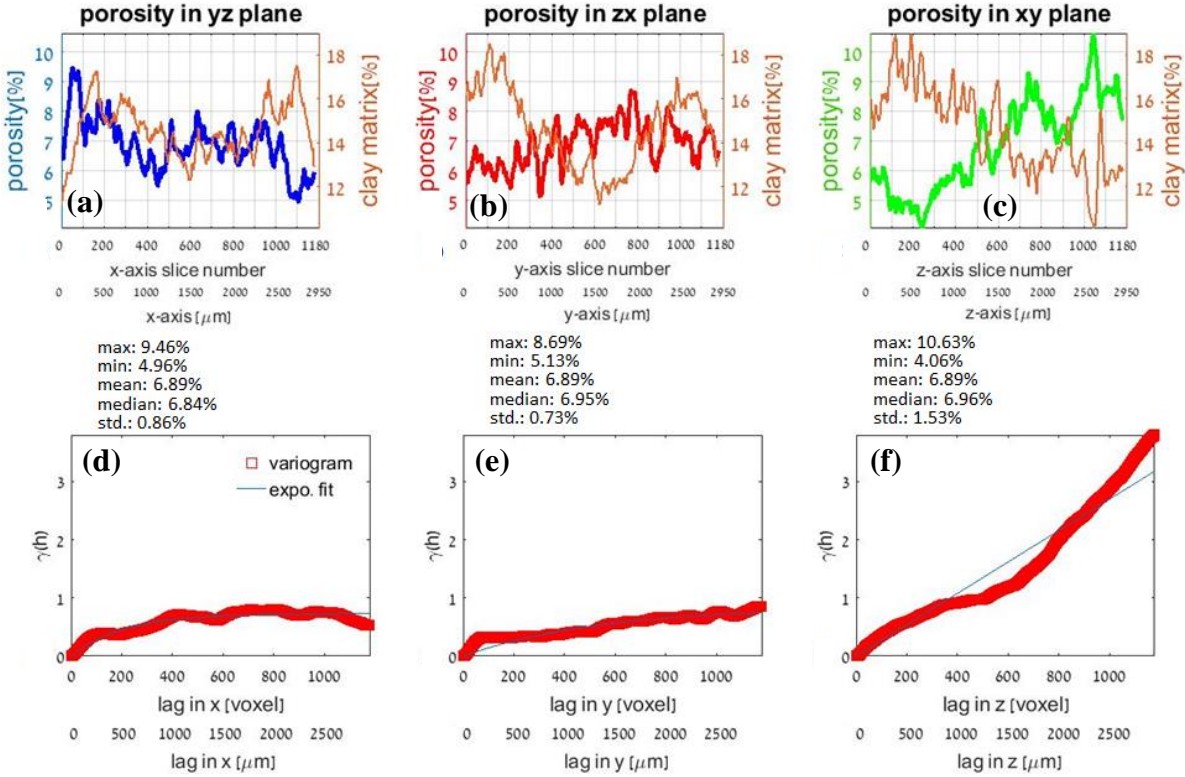

**Figure 13.** *Advanced REV analysis for Sample S2. In the top row porosity calculated slice-by-slice for x-, y- and z- directions and fraction of clay matrix phase in the image (**a-c**) are presented. At the bottom row (**d-f**) conducted variogram analysis indicates that representativeness in x-direction is reached for the large cube edge size of 500 voxel (1250 μm), but for y- and z- direction it is not reached at all and therefore REV could not be specified in the CT-scanned sample S2.*

Sample S3 (Fig. 14): In this CT specimen, all three directions show similar fluctuations around the mean porosity (Fig. 14a-c), as expected from the ordered distribution of pores (see also Fig. 7). The difference between the minimal and maximal IPs is 5.89 % in y-direction, and 5.45 % for the x- and z-directions. Standard deviation is largest for y-direction but it does not differ significantly from that in x- and z-directions. Also, median for each direction shows very close values to the mean.

Variogram analysis (Fig. 14d-f) indicates homogeneity of the sample for relatively small sub-volumes. Representative size (the variogram range) is attained at ~100 voxel (250 μm) cube edge size and therefore



REV of this size could be assumed. However, because it is the size of two average grains only, REV in S3 is
defined by the classical analysis as a cube of 350 voxel (875 µm) edge size.

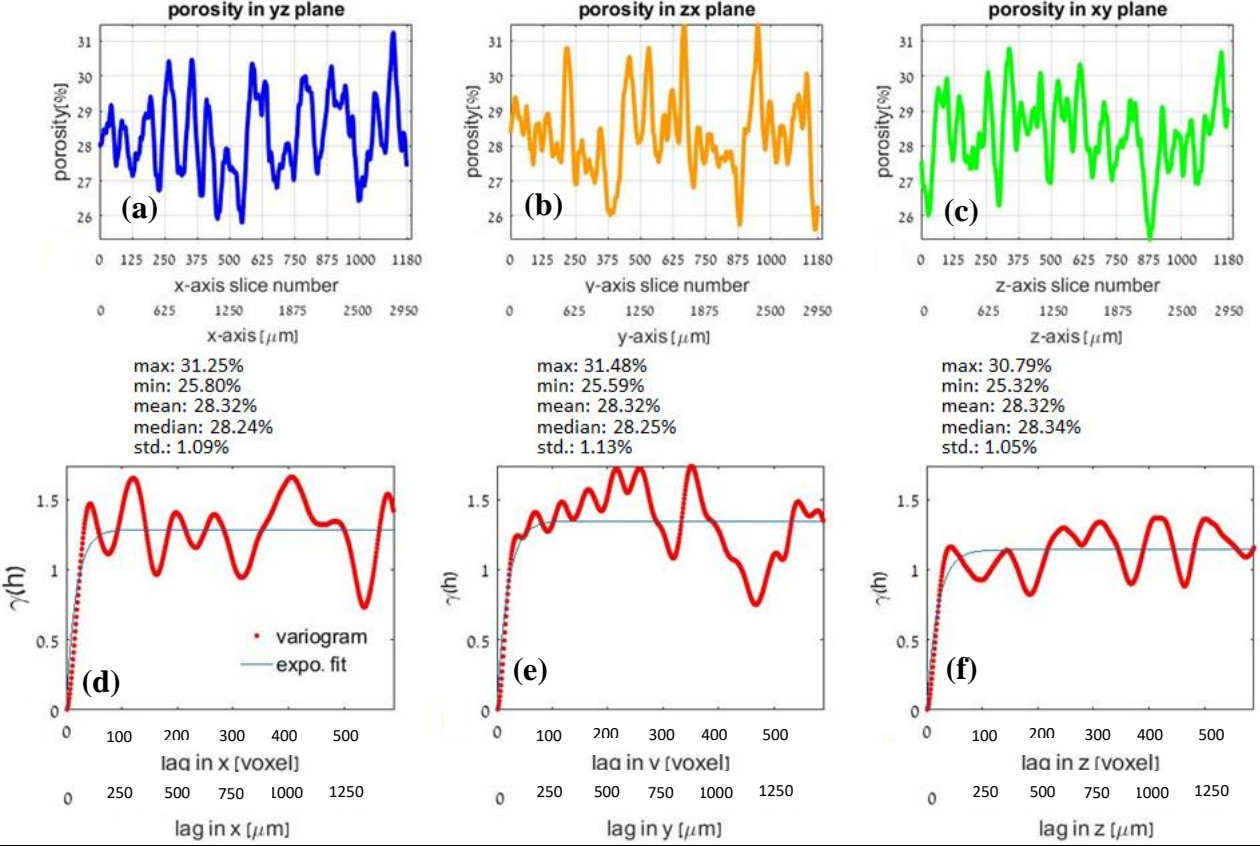


***Figure 14.*** *Advanced REV analysis for Sample S3. In the top row porosity calculated slice-by-slice for x-, y-*
*and z- directions is presented (**a-c**). At the bottom row (**d-f**) conducted variogram analysis indicates*
*representativeness of the sample for relatively small sub-volumes of 100 voxel (250 µm) cube edge size.*
**3.3. Fluid flow modelling at a micro-scale**
Samples for modelling: Creeping Flow (Sect. 2.2) was modelled at the pore scale in two µ-CT-scanned
geometries: 1) Full Sample S1 of 1180 voxels (2950 µm) size, including two adjacent parts of lower and
higher porosities, and 2) Sample S3 REV of 350 voxels size (875 µm). Modelling in the Sample S2 was not
performed due to the reasons detailed above.



Pressure difference between the inlet and outlet boundaries was prescribed each time for three

orthogonal directions to produce a steady-state velocity field (a constant pressure gradient of 2.424 $\left[\frac{Pa}{mm}\right]$ was
used in all the simulations for consistency).

***Table 3.*** *Porosity loss in three samples in a course of application of the extended computational*

*workflow.*

| Sample | Sample size (mesh size) [μm] | Total volume [$\cdot 10^9 \mu m^3$] | CT segmented image | Connected porosity | Mesh porosity | | | Gas porosity, % |
|---|---|---|---|---|---|---|---|---|
| | | | Porosity, % | Porosity, % | Porosity, % | Pore Surface area [$\cdot 10^6 \mu m^2$] | Specific surface area (SSA) [$\mu m^{-1}$] | |
| S1 (entire sample 1180 voxels) | 2950 (14) | 25.67 | 17.52 | 15.63 | 13.6 | 186.7 | 0.053 | 28 |
| S3 (REV 350 voxels) | 875 (5) | 0.670 | 28.32 | 27.96 | 25.93 | 11.13 | 0.064 | 31 |


Sample 1 – full sample 1180 voxels (2950 μm):

This specimen includes two different regions of low (0-250 voxels) and higher porosity (250-1180

voxels) in z-direction (Fig. 11c). Porosity of the meshed domain is 13.6 %, compared to 17.52 % in the
segmented image (Table 3). Mesh edge size on the pore walls is 14 μm. max $Re = 0.084$ assured the creeping
flow regime. Calculated permeability tensor, $\bar{\bar{\kappa}}$ (Eq. (5)) was symmetrised (Eq. (6), Table 2):
$$\bar{\bar{\kappa}}_{sym} = \begin{pmatrix} 420 & 66.3 & 1.91 \\ 66.3 & 344 & 12.8 \\ 1.91 & 12.8 & 163 \end{pmatrix} \qquad (7)$$



It agrees with the variogram analysis (Fig. 12f), which shows a higher variance for porosity in the z-
direction, because of a cementation presented by horizontal (x-y plane) laminas (Fig. 3).
Sample 3 – REV 350 voxels (875 µm):
Porosity of the meshed domain is 25.93 %, compared to 28.32 % in the segmented image (Table 3).
Mesh edge size is 5 µm on pore walls. Maximal $Re = 0.22$ assured creeping flow regime. The symmetrised
permeability tensor is close to isotropic (Table 2):
$$\overline{\overline{\boldsymbol{\kappa}}}_{sym} = \begin{pmatrix} 4517 & 5 & 38 \\ 5 & 4808 & 547 \\ 38 & 547 & 4085 \end{pmatrix} \tag{8}$$
For S3 an average tortuosity in x, y, z directions (calculated with a particle tracing tool of Comsol
Multiphysics) varied in the range [1.39, 1.47] (Table 2), with lowest value associated with the largest
permeability in y-direction, and the largest value associated with the smallest permeability in z-direction, as
expected.
**3.4. Image analysis**
Image analysis (Sect. 2.2) was performed on a segmented image of the whole sample of each specimen,
i.e. on a cube of 1180 voxels edge length (2950 µm) scanned with resolution of 2.5 µm. Pores were separated,
while those touching the external boundaries were excluded.
Sample S1: The mode peak of pore size distribution (measured by pore Feret maximum calliper) (Fig.
15) is at 194 µm with FWHM at [150,335] (Table 2). In total 3500 pores were analysed.






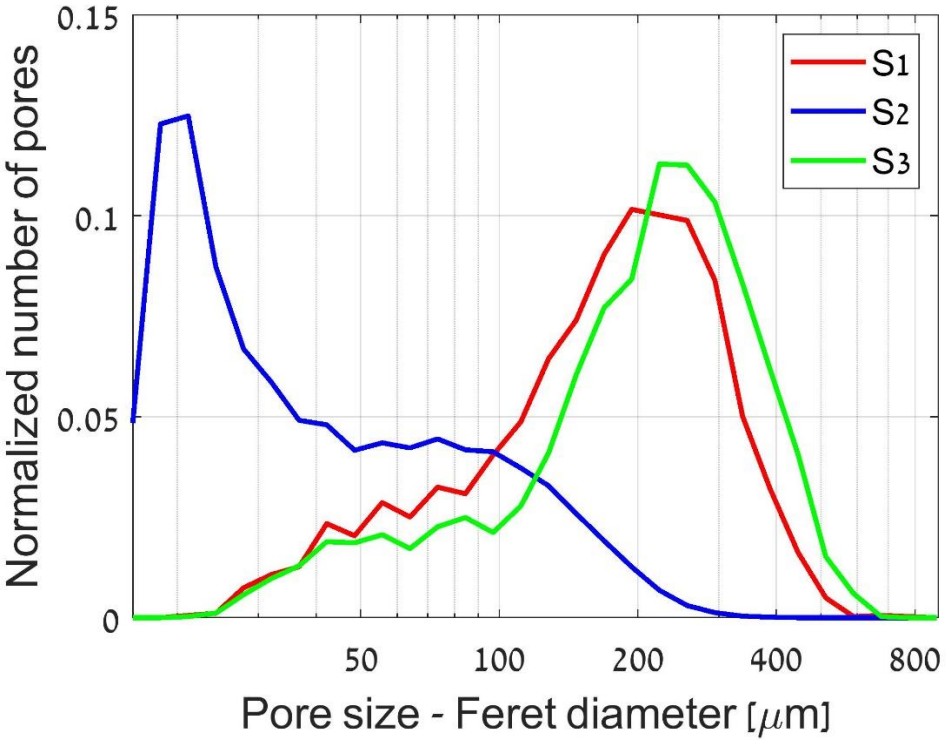


***Figure 15.*** *Statistics of pores calculated from the image analysis. Sample S1: Pore size distribution has a*
*peak at 194 µm with FWHM at [150,335] and shows a Gaussian shape. A large size of pores population is*
*recognized at ~60 µm as well presenting the pore throats. Sample S2: Pore size distribution has a peak is at*
*21 µm and does not show a Gaussian shape. A large size of pores population is recognized at ~100 µm as*
*well. Sample S3: Pore size distribution has a peak at 223 µm, with FWHM at [145,400] µm and shows a*
*Gaussian shape. A significant pore population is recognized at ~50 µm as well presenting the pore throats.*
*X-axis is logarithmic.*
Specific surface area (SSA) calculated from µ-CT images is $0.068\,\mu m^{-1}$ (Table 2) . Average
tortuosity, $\boldsymbol{\tau}$, measured on the whole CT image by multiple paths indicates close values in x and y directions,
1.37 and 1.38, correspondingly, whereas in the z-direction it is 1.48 (Table 2). As many paths were considered,
we suggest that the difference is created by the textural features that appear in horizontal planes (Fig. 3a).





Sample S2: The mode peak of pore size distribution (Fig. 15) is at 21 µm, while the curve does not
show a gaussian shape. A large pore population is recognized also at ~100 µm (Table 2). In total 45000 pores
were analysed. Specific surface area (SSA) calculated from µ-CT images at resolution size is
0.136 $\mu m^{-1}$ (Table 2), which is twice larger than that of S1.
Sample S3: Pore size distribution (Fig. 15) has a peak at 223 µm, with FWHM at [145,400] µm and
shows a gaussian shape (Table 2). In total 3491 pores were analysed. An average tortuosity, $\tau$, measured on
the whole CT image with multiple paths is 1.32, 1.34 and 1.39 in the x-, y- and z-directions, respectively. It is
seen that this geometry-based tortuosity, $\tau$ (Table 2), is lower for S3 than for S1 in all directions, because S3
has less cement at pore throats.

## 627   4.   Discussion

### 628     4.1. Hatira Formation geological characteristics

The three sandstone samples of the Hatira Formation at Ein Kinya sandstone explored in this study
show characteristics similar to those of the Kurnub Group – Hatira Formation elsewhere in Lebanon, Jordan
and Israel (Massad, 1976; Abed, 1982; Kolodner et al., 2009). Their main features are: textural maturity, grain
roundness, and sorting, Figs. 3-5. Mineralogical maturity indicated by the dominance of quartz and very small
proportion of feldspars, kaolinite, as the only clay minerals detected by X-ray diffraction. These features
suggest a redeposition of Palaeozoic Nubian Sandstones (Kolodner, 2009). The Fe-oxide can be also derived
from the original Palaeozoic Nubian sandstones as coatings of the quartz grains and as detrital Fe-ox grains.
As fossils and carbonate minerals were not detected, whilst cross bedding, graded bedding, and interbedding
of a horizon enriched in silt and clay between the quartz arenite, may suggest a fluvial environment of
deposition.
The top (S1) and bottom (S3) layers (Fig. 1c) are classified as quartz arenite with good sorting and small
extent of fines, separated by 20 cm thick quartz wacke sandstone layer (S2) poorly sorted with 34 % of fines,
which form the clay matrix. Despite the differences in grain size distributions, the three layers show similar
grain size populations with different weights of each population (Fig. 6): main population of a fine-medium
sand, and smaller weights of coarse silt (~40 µm) and clay (~1 µm). Therefore, it is suggested that the source
sediments have arrived from the same provenances.



The top sandstone layer (S1) (Fig. 1) is characterized by Fe-ox grain coating and meniscus type cementation (Fig. 3). The intermediate sandstone layer (S2) contains clay-matrix with Fe-ox cementation (Fig. 4) and therefore has low permeability. The bottom sandstone layer (S3) has clean quartz grains with very low extent of co-occurring Fe-ox cementation (Fig. 5). With no features indicative of a marine environment features, the grain coating and meniscus cement derive their occurrence at the partly saturated conditions of meteoric water (Worden and Burley, 2003). The extent of iron oxide cement depends on supply of its reactants, i.e. iron and oxygen. Under the unconfined-phreatic conditions, meteoric water infiltrates the rock and supplies the iron solute. Oxygen is available from the atmosphere and from the infiltrating water promoting oxygenated condition, where the iron is the limiting factor on Fe-ox precipitation.

Local patches of Fe-ox grain coating and meniscus type cementation at the scale of sub-mm to a few mm's in the top layer sample S1 are associated with exceptional large quartz grains, located at highly permeable regions (Fig. 3b), where preferential paths of fluid are abundant. These paths of meteoric water supplied dissolved iron that resulted in iron oxide cementation, where the oxygen was supplied either by the meteoric water or by infiltration of air through the partly-saturated realm conditions. The non-uniform cementation pattern at Darcy scale (mm's to cm's) is a result of hydrodynamic dispersion within heterogeneous porous medium. The yellow-brown colour is associated with a goethite mineral cement.

The bottom layer S3, which has a texture similar to that of the top layer S1, is overlain by the less permeable intermediate layer S2, which controls the conditions in the unsaturated zone. A small amount of meteoric water infiltrated the bottom layer, causing low amount of iron supply, and resulting in low supply of iron, which resulted in low iron oxide meniscus cement and grain coating. Low cementation lead to poor consolidation. The reddish colour of the bottom layer suggests the presence of a hematite mineral cement. The very small amount of fines observed (Fig. 6), suggests a small contribution of suspended clay through the vadose zone. A possible source for the clay (0.8 %) is an observed pressure solution.

Sample S3 rock is characterized by a pattern of sub-mm-scale parallel bands of reddish colour due to different extent of iron oxide (Fig. 5b). This pattern may represent "Liesegang bands" (Liesegang, 1896), zones of authigenic minerals (iron oxide in this case) arranged in a regular repeating pattern. The banding pattern indicates precipitation at a chemical interface that favours iron oxides precipitation (Foos, 2003). This could be a redox front, pH front, associated with saturated water level interface, which changes with time.





Increasing acidity favours dissolution of $Fe^{3+}$ at positive redox potential (Eh) values. As Eh decrease, the Fe-
ox precipitated again on the grains at the interface with the saturated water zone. The limiting factor is assumed
to be the oxygen, with lower concentration below the water table.

In the intermediate layer S2 the grain sorting is poor. The prominence of clay matrix is indicated by

horizontal layering. On a microscopical scale the clay forms point contacts (bridging) between detrital sand
grains. These imply that the clay is not post depositional to the sand. Poor vertical sorting in the layer indicates
changes in deposition energies. The abundance of fines indicates conditions of slow water flow. The grey-
white colour of the rock may reflect the abundance of kaolinite.

During the diagenetic compaction stage, the high porosity of both top and bottom layers (Table 2) has

been preserved due to meniscus cement consolidation (Figs. 3, 5). However, grains experienced pressure
solution indicated by concave-convex contacts and mutual inter-penetration (Fig. 5d), along with mechanical
breakage and cracking. In contrast, under the burial loading the intermediate layer experienced compression
and compaction of clays to agglomerates (Fig. 4).
**4.2. Influence of pore network microscopic characteristics on permeability**

Each of the evaluated micro-scale rock properties can supply qualitative information about the macro-

scale permeability (Tables 2, 4). Intrusion of mercury (effective saturation) with increasing pressure shows
similar slope in samples S1 and S3 (Fig. 7a), suggesting their similar structural connectivity at the macro scale.
Lower threshold pressure in S1 is due to larger grain size, and its lower saturation is due to larger extent of
fines compared to S3. For S2, no threshold pressure is a result of fines filling the inter-granular space
sporadically.
**4.2.1. Pore and pore throat size distribution**

Gas porosity of the bottom layer (S3) is slightly higher than that of the top layer (S1) (31 % vs. 28 %,

respectively, both are quartz arenites, Table 2), because of the larger extent of infiltrating and deposited fines
and more cementation at the top layer, reducing the pore space. Analysed by mercury intrusion porosimetry,
the volume fraction of the pore space that is controlled by bottle-necks of macro pore-throats (larger than 10
µm), was 93% for the bottom layer, and 81% for the top layer (Fig. 7), suggesting that fines reduced the pore
throat size. The skewness of the pore throat size distribution of the top layer indicates an increase in the amount
of fines resulting in the reduction of the effective pore space available for fluid flow, and increase in its



heterogeneity. The intermediate layer (S2) comprises more fines, which form clay matrix with 19 % of
porosity (Table 2) under the burial conditions. Only ~10% of the pore space volume fraction is controlled by
bottle-neck macro pore-throats (Fig. 7). All three layers presented the same pore size populations, with
different extents. The top and bottom layers are characterized by primary pore throat mode of 44 µm and 35
µm with narrow distribution (Table 2, Fig. 7), and pore size distribution mode values of 194 µm and 223 µm,
correspondingly (Table 2). For the intermediate layer the intergranular porosity was distributed over a wide
range: from ~1 µm of pore sizes reduced by fines, to a very few pore throats as large as ~40 µm, where less
clays deposited or infiltrated into. The secondary population of the pore throat size for the top layer is focused
around 35 nm (Table 2, Fig. 7b) (8 % of pore space), presented by pore throats between the iron oxides flakes.
The bottom layer presents this population in tiny amounts due to little iron oxides cementation. In contrast,
the intermediate layer presents a large extent of this population (40 % of pore space), associated with pores
between both the iron oxides flakes and inside the clay matrix. In addition, 3D image analysis of pore size
distribution in the intermediate layer indicated a primary pore size mode of 21 µm and a secondary pore size
mode of 100 µm (Fig. 15b, Table 2).

The characteristic length of a porous rock has a similar size to the main pore throat mode for the top

and bottom layers (42.9 µm and 36.9 µm, correspondingly, Table 2), which are both characterized by sorted
pore size distribution (Fig. 15). In contrast, the intermediate layer, characterized by poor pore throat size
distribution, had a characteristic length of 12.3 µm, where only 8% of the pore space volume is controlled by
these bottle-neck pore-throats (Fig. 8). It shows that even when pore space includes mainly a sub-micro-scale
porosity, porosity type controlling the flow may still be attributed to the macro-pores.

In addition, pore throat length contributing to maximal conductance, $l_{max}$ (Fig. 9), indicates the

optimum path for flow at increasing pressure. This is of a special interest when extracting subsurface fluids.
For all three investigated samples this pore-throat size is smaller than characteristic length (Fig. 8, Table 2),
when the relative decrease is greater for the layers containing more fines.
**4.2.2. Grain roughness**

Segmented CT image porosity (IP) is limited by the image resolution of 2.5 µm, and thus should be

lower compared to the experimental porosity (Tables 2, 3). The difference between IP and CT predicted image
porosity from MIP (Table 3) may be used to assess grain coating and surface roughness. µm-scale cement



coating (including Fe-ox flakes separated by voids) is usually erroneously assigned on segmentation to grains
rather than to pores, due to a partial volume effect (Cnudde and Boone, 2013). This is because X-ray
attenuation of Fe-ox is higher than of $SiO_2$, which generates voxels of high intensity (Lide, 2003). Hence,
surface roughness can be quantified by the ratio between IP and CT predicted image porosity from MIP (which
in our case have closer values for the clean sample S3 rather than for cemented S1, Tables 2, 3). Therefore,
image pre-processing steps (image processing and segmentation) should be performed with high precision and
caution.

Pore surface roughness may be evaluated from the specific surface area (SSA- surface-to- bulk-volume)

measured by MIP, which considers pores larger than 0.006 μm (Table 2). The larger SSA implies a rougher
surface (e.g. Tatomir et al., 2016). SSA for samples S1 and S2 (3.2 $\mu m^{-1}$ and 12.2 $\mu m^{-1}$, respectively) are
similar to those given in the literature for sandstones of similar properties (e.g. Cerepi et al., 2002). The SSA
value of Sample S2 is higher because of its high silt and clay content of 34.3%, which is 7.4% only for S1
(Fig. 6a). SSA of Sample S3 (where silt and clay constitute 5.6 %, including Fe-ox rim coating) is 0.16 $\mu m^{-1}$
only, which is 20 times smaller than SSA of Sample S1 (Table 2).

**4.2.3. Connectivity index**

Connectivity index (Eq. (7)) of S3 (10) is about three times higher than that of S1 (3.49) (Table 2)

because some of the pore throats of S1 were clogged by cement and fines. Sample S1 has a lower connectivity
than it could be expected from well-sorted sandstone, which is also indicated by the lower value of IP (17.52
%, Table 3) compared to CT porosity predicted from MIP (23.5 %, Table 2), due to the partial volume effect
at grain boundaries discussed above.

In the quartz wacke of Sample S2 CT specimen, Euler characteristics, χ, was calculated as a sum of χs

in a cluster of a main pore network, and that in a few smaller ones. The connectivity index of S2 (0.94, Table
2) is lower than that of both S1 (3.49) and S3 (10), because of the clay matrix which clogs pores. It is important
to mention that Euler characteristic depends on image resolution: smaller pixel size would reveal smaller pores
and more connections and assure a quality of resolution, whereas larger pixels may be assigned as "grain"
(estimated through the inversion with a higher image intensity value) and block the pores connection.





In summary, although S1 pore network has larger pore throats, it also has larger grain roughness, and
lower connectivity compared to S3. The two latter properties dominate and generate a smaller permeability of
quartz arenite sandstone S1 compared to S3 (see permeability tensor, Table 2).
**4.3. Empirical approximations of permeability: Connections between micro- and macro-scale**
**rock properties**
Macroscopic permeability can also be approximated by some empirical and analytical relations,
involving microscopic and macroscopic rock properties measured in this study (Table 2). These approaches
started with Kozeny (1927) and Carman (1937) but their challenging goals have not been completely achieved
yet.
For instance, permeability can be approximated using Kozeny-Carman equation (Bear, 1988):
$$\kappa = c_0 \frac{1}{\tau^2} \cdot \frac{\phi^3}{M_s^2 (1-\phi)^2} \qquad (8)$$
where $c_0$ is a coefficient called Kozeny's constant, varies according to the geometrical shape of the
channels (for equilateral triangle pore $c_0 = 0.597$, Bear, 1988), $\tau$ is the tortuosity (measured from the μ-CT
data for samples S1 and S2 only, Table 2), and $M_s$ is the specific surface area (including the micro pores,
Cerepi et al., 2002), calculated relative to the unit volume of solid ($M_s = \frac{SSA}{(1-\phi)}$; SSA is the specific surface
area scaled with the bulk volume of the sample and evaluated from MIP). As it was impossible to evaluate
tortuosity in Sample S2, therefore $c = c_0 \cdot \frac{1}{\tau^2} = 0.2$ reported by Carman (1938) was used to fit the experimental
data. For S1 and S2, results show (Table 4) that approximation by Kozeny-Carman equation gives slightly
higher permeability relative to the direct experimental measurements. For S3, Kozeny-Carman permeability
is ten-fold larger than the lab permeability in z-direction, and at the same scale with that in x-y plane, thus
showing isotropy. The suggested reason for the difference in permeability between S1 and S3 in Kozeny-
Carman approximation is the accounting for the specific surface area, which for S1 is larger because of poorer
grain sorting and larger extent of Fe-ox cement flakes at the grain surface.
An empirical relation between permeability, porosity, and a capillary pressure parameter is presented
by Winland's equation (Winland, 1976; Pittman, 1992; Kolodzie, 1980) based on laboratory measurements of
mercury intrusion:





$$\log r_{35(\mu m)} = 0.732 + 0.588\log\kappa_{(mD)} - 0.8641\log\phi_{(\%)}, \tag{9}$$
where $r_{35(\mu m)}$ is the pore throat *radius* at 35 % mercury saturation, defined as a function of both pore
throat entry size and sorting, serving as a good measure of the largest connected pore throats in a rock
(Hartmann and Coalson, 1990). This permeability estimation (Table 4) yielded doubled values for S1 (with
respect to the experimental measurements, Table 2), and some average values of horizontal and vertical
permeability for Samples S2 and S3.
Permeability as a function of pore size and porosity (Katz and Thompson, 1986) can be approximated
as:
$$\kappa(l, \phi) \approx 4.48 l_c^2 \phi^2, \tag{10}$$
where $l_c$ ($\mu m$) is the characteristic length of the pore space (Table 2, Fig. 8). The results agree with
those from the lab measurements for Sample S1, slightly overestimate those for S2, and for S3 suite better the
vertical permeabilty rather than the horizontal one. Results calculated from Katz and Thompson (1987) (Eq.
(1)) based on $l_c$ and $l_{max}$ are presented in the Table 4 as well.
***Table 4****: Empirical approximations of permeability. Permeability approximated by different methods*
*explained in the text is presented and compared to the permeability from the flow modelling and from gas*
*permeametry (Table 2) (presented in the two last rows).*

| | $\kappa_{S1}$ [mD] | $\kappa_{S2}$ [mD] | $\kappa_{S3}$ [mD] |
|---|---|---|---|
| Kozeny-Carman (Eq. (9)) | ⊥ 526 <br> ‖ 598, 608 | 8.1 | ⊥ 3575 <br> ‖ 4050, 3880 |
| Winland's equation (Eq. (10)) | 1325 | 4.5 | 1790 |
| Katz and Thompson 1986 (Eq. (11)) | 617 | 12.3 | 658 |
| Katz and Thompson 1987 (Eq. (1)) | 330 | 4 | 460 |
| Flow modelling (Table 2) | ⊥ 163 <br> ‖ 344, 420 | - | ⊥ 4085 <br> ‖ 4808, 4517 |



| Gas permeability (direct experiment) (Table 2) | ⊥ 350 || 640 | ⊥ 2.77 || 7.73 | ⊥ 220 || 4600 |
|---|---|---|---|

**4.4. Upscaling permeability: accuracy of the extended computational workflow**

Permeability was upscaled in our study by averaging over the fluid velocity field (Eq.(4)) calculated by free-flow modelling at the real geometry in the REV sample. Therefore, each step in this extended computational workflow (Fig. 2) affects the upscaled permeability.

**REV determination**: REV was determined by two approaches – by the classical and directional techniques. Initially, REV analysis was conducted inside a search domain of 3 mm cube. For S1, classical method determined REV cube of 1187 μm edge (Fig. A1 in the Appendix A). Variogram-based directional method yields 250 μm REV size in x-y plane, while 875 μm in z-direction (Fig. 11). Also, the variogram sill, which refers to the variance of the slice-by-slice porosity was ~3 times larger in z- direction than those in x- and y- directions.

In addition, slice-by-slice porosity in the bottom 625 μm of the specimen S1 is lower by ~3 % than in the top specimen's part (Fig. 11c). The lower porosity in that section may be associated with a higher amount of cement (orange curve in Fig. 16). However, it differs between the parts by 0.5-0.75 % only, which is smaller than 3 % difference in porosity between the same parts (Fig. 11c). We suggest that since the sizes of iron oxide cement flakes are at the scale of resolution (2.5 μm), the amount of the imaged cement may be underestimated.

Moreover, correlation was found between the grain size (measured also by image analysis tools) and cement (Fig. 16) that can also be observed in the thin section presented in Fig. 3b. Near the cemented region at ~750 μm, exceptional large grains are found (Fig. 16, indicated by red rectangle), brought probably by some higher energy depositional event. Large grains cause large pores and generate more permeable horizons, where water flow was presumably focused (McKay et al., 1995), supplying iron solutes. We suggest that after the flooding events a vadose zone formed, where a dominant water flow mechanism changed from gravitational to capillary one. Then, water flowed due to capillary forces along grain surfaces towards regions with larger surface area and iron precipitated in a reaction with oxygen available at the partly saturated zone. We suggest that over time this cementation mechanism caused decrease of throat size nearby the preferential path, while the preferential path itself with the large pores remained open, eventually generating anisotropic flow pattern.

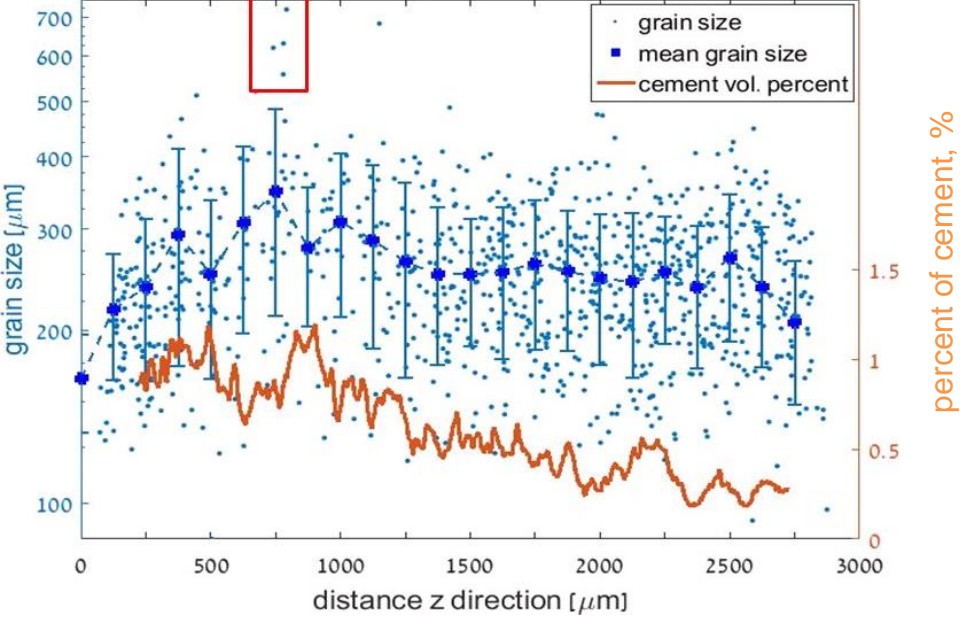

**Figure 16.** *Grain size scattering and Fe-ox cementation (left and right vertical axes) in sample S1 in slices in z-direction. Size of grains is indicated by blue dots, mean grain size in is indicated by blue circle, and percent of Fe-ox cementation is shown by orange line.*

Finally, additional variogram analysis conducted in a volume of 7-9 mm edge size of S1 (scanned with resolution of 5 μm) shows a larger-scale cyclicity in z- direction (in contrast to that in x-y plane) (Fig. 12c-f) due to repetition of lower porosity bands at 2 mm distance indicated by the maximal correlation length. Therefore, two scales of variation can be derived from the variogram analysis of the sample S1: fluctuations at 300's μm correlation length due to size variability of grains and pores (Fig. 11f), and the fluctuation at 2 mm correlation length due to the appearance of the higher and lower porosity bands explained above (Fig. 12f). The larger-scale type of variability can be inferred also from the classical REV analysis (Fig. A1 in the Appendix A). Specifically, mean porosity lower than the median one, points on a larger-scale heterogeneous feature with porosity lower than the homogenous field of investigation. This statistical analysis can be used as an indicator for a larger-scale heterogeneity feature in the sample. Due to this large correlation length, in S1 we used the whole specimen cube volume of 2950 μm edge at 2.5 μm resolution for the flow modelling.

For S3, the classical REV-method determined REV cube of 875 μm edge, where disagreement between the mean and median became very small (Fig. A3 in the Appendix A). Directional method determined REV



of 250 μm (Fig. 14). This length is equivalent to two grains diameters, presenting high homogeneity in the
specimen 3mm cube edge. The larger REV from both approaches was chosen (Table 3).

For S2, both REV methods indicated REV size larger than the sample size (Fig.A2 in the Appendix A,

Fig. 13). Mean porosity larger than the median points on larger-scale of heterogeneity feature with higher
porosity, possibly larger inter-granular pores with less fill of clay matrix (Fig. 13). Fig. 13a-c show slice-by-
slice porosity along with clay matrix (presented by the brown curve). Average of clay matrix was 15.5%,
where mean porosity from μ-CT images was 6.89 % (at 2.5 μm resolution limit). Inverse correlation of
porosity and clay matrix is identified, most distinctive in the z-direction. In that direction porosity has an
increasing trend, and therefore there variogram has no sill (Fig. 13f), where in the x-direction the sill converges
to the value smaller than 1 (Fig. 13d). This observation indicates a prominent anisotropy. The higher variance
of porosity and clay matrix in z-direction means that the clay matrix pattern is related to horizontal mm-scale
layering. For those reasons the analytical program formulated in our paper can't be entirely applied to sample
S2, due to impossibility to determine REV and to conduct subsequent pore-scale flow modelling. As a result,
although sample S2 presents a common sandstone, its heterogeneous nature and anisotropy allow conducting
the experimental measurements only.

**A source of the inaccuracy is the use of porosity REV for the permeability measurements**:

Mostaghimi et al. (2013) showed for sandpacks that the REV cube edge for permeability is twice larger than
that for porosity, where the ratio increases with sample heterogeneity. The latter relies also on contributions
of tortuosity and connectivity of the pore space. These components add another uncertainty to determination
of the upscaled permeability.

**Imaging**: The CT image resolution of 2.5 μm limits the reliability of presentation of the porous medium

and defines the lower limit for pore identification using this method. As explained in the methods section, we
applied an image processing and segmentation workflow to recover the image geometry, which was blurred
by noise or affected by partial volume effect. Then, we estimated the loss of pore space due to the resolution
limits by the amount of mercury which filled the pore space in the MIP experiment. After segmentation,
sample S1 had porosity of 17.5 %, and 23.5 % for the CT porosity estimated from MIP (Tables 2, 3). In this
sample grain coating flakes of iron oxide with high attenuation coefficient were common, growing also on top
of each other to the size of tens of microns (Fig. 3). Therefore, the difference in porosities generated by the




partial volume effect is a significant component of the error, especially for the tiny structures, such as pores,
with a large ratio of surface to volume (Kerckhofs et al., 2008). Porosity of S3 after segmentation (IP) was
28.3 %, which is close to 30.4 % estimated from MIP (Tables 2, 3). This is a result of the very small degree
of cementation and the absence of iron oxide flakes in the majority of the sample, leading to the small
contribution of the partial volume effect. The IP value of S2 was 6.89 %, where the estimated porosity from
MIP was 6.65 % (Table 2). There is a clay cover on grains (in addition to the clay matrix) in this sample,
which is supposed to lead to the lower IP than porosity estimated by MIP. However, in mercury porosimetry,
large internal pores are not filled, unless the pressure is sufficient to fill a pathway towards these pores. This
causes a bias in the pore throat distribution curve (Fig. 7) towards the smaller pore throats. In addition, porosity
estimated by MIP at resolution limit is sensitive to the volume of mercury intruded due to change in the
mercury pressure (the slope of the curve in Fig. 7a at the intersection with red dashed line). The large slope of
S2 saturation curve at the resolution limit (which is much larger than those for S1 and S3) introduces
uncertainty to the porosity estimated by CT. Moreover, the reliability of this method would be higher for well-
connected porous media.
**Identification of connected domains**: The geometry used in the fluid model included only the pore
network that connected the six faces of the REV cube. Other pore space in the REV, which was disconnected
from the main network (partially because all paths to them were assigned to grain pixels due to the partial
volume), was deleted, thus resulting in the smaller size of the simulation domain. IP of sample S1 was 17.52
%, whereas its connected porosity was estimated as 15.63 % (Table 3). Those of sample S3 were 28.32 % and
27.96 %, respectively (Table 3). The larger decrease in connected porosity of S1 is related to the decrease in
pore throats due to higher abundance of fines and to iron oxide cementation. In contrast the, connectivity of
S2 is determined by the numerous finer porosity networks disconnected from each other, due to the high
amount of clay matrix.
**Transformation of geometry bitmap images to grid mesh**: Mesh was generated considering a trade-
off between the size of the mesh elements (4 elements in the smallest pore throat) and computational limits,
while coarsening the mesh elements toward the pore centre. Connectivity between the pores with very fine
pore throats that could not be replaced by the mesh element could be lost, which resulted in a loss of those
pores. For instance, in the transformation to mesh S1 connected porosity was reduced from 15.63 % to 13.6
% of the mesh porosity, whereas gas porosity was 27.4 % (Table 3). Therefore, porosity used in simulation



was 50% smaller than porosity estimated by gas porosimeter. This loss of porosity is supposed to introduce a
significant uncertainty in permeability estimates, upscaled from the velocity field in the specimen. The
connected porosity of S3 was reduced from 27.96 % to 25.93 % of mesh porosity, where the porosity estimated
by gas porosimeter was 31 %. Therefore, porosity used in simulation was mostly preserved, comprising about
84% of that estimated in the lab.
**Permeability tensor:** The permeability tensor of S1 has the lowest value of permeability in z-direction,
163 mD, and the highest in x-direction, 420 mD (showing ~3 times difference). The anisotropic permeability
is explained by the lower porosity in the bottom part of the specimen (Fig. 11c), where the restriction to flow
was due to higher degree of cementation. The observation on a larger volume shows cyclicity (Fig. 12f) with
maximal correlation length of 2 mm (discussed above). The loss of magnitude of permeability in our results
(z-direction compared to x-direction) is due to the 50 % loss of the porosity, part of it of macro-pore
transmissive medium. Permeability anisotropy trend is in agreement with the variogram analysis which
showed larger sill and range in the z- direction (Fig. 12f). For comparison, Clavaud et al. (2008) calculated
permeability in a saline tracer test using X-Ray imaging in clay-free sandstones and obtained permeability in
x-y plane in average twice larger than that in z-direction. This effect was related to the presence of less
permeable silty layers. Their degree of anisotropy is smaller than in our results. This effect is explained in our
case by the loss of connecting pore throats close to the specimen faces, which change the simulated flow
pattern and the calculated flux through each face.
MIP permeability measured at a larger scale (in a cube of 3 mm edge) was 330 mD (Table 2). In gas
permeameter of cylindrical sample with ~5-7 cm height and 2.5 cm diameter, permeabilities were 350 mD
and 640 mD, in z-direction and x-y plane, respectively. The flow modelling predicted successfully the
permeability anisotropy (discussed above). It was lower than that determined with a permeameter, and is, on
average the same as that resulted from MIP. The 50 % loss of porosity in the simulated specimen in comparison
to the real sample is assumed to cause the lower permeability resulting from the flow modelling.
Permeability measured with a gas permeameter yielded ~4600 mD in the x-y plane, whereas 220 mD
only in z-direction (Table 2). We suggest that when gas was flowing through the poorly consolidated Sample
S3, grains could be dislodged from the bulk sample, mostly affecting the measurements conducted on the x-y
plane, parallel to Liesegang cementation bands, which were observed in a thin section (Fig. 5c). These bands

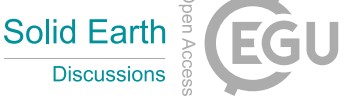


show horizontal cementation at sub mm-scale, which may restrict the flow in z-direction. Missing grains could
produce "tunnels" resulting in high flux of gas thus generating a high permeability value in the x-y plane.
Alternatively, the observed 20-time difference in permeabilities could be explained by a significant effect of
the slightly lower-porosity banding in the x-y plane on the corresponding permeability in z-direction. This
could also be inferred from the empirical approximations, e.g. $\kappa \sim \phi^n$, Eqs. (8,10). Similar phenomena were
reported in other studies. Permeability measured in sandstone samples by Oyanyan and Ideozu (2016) in x-y
plane was at most 1.5 times larger than that in z-direction, whereas the largest anisotropy was in mud drapes
lithofacies with maximal anisotropy reaching a factor of 6.

The modelled permeability tensor of our sample S3 in x-y and z-directions resulted in approximately

isotropic values, ~4500 mD (Table 2). This isotropic behaviour is in agreement with the similar variogram
sills and ranges in the directional REV analysis in all the three directions (Fig.14). Permeability derived from
MIP on 1cm size sample was 466 mD, i.e. ten times lower than the simulated one. Therefore, we suggest that
the sample for the CT imaging and flow modelling was retrieved from the higher-porosity regions of the
macroscopic sample. In addition, permeability values from μ-CT flow modelling obtained by Tatomir et al.
(2016) on a similar sandstone exceeded gas permeability by ~6 times for the fine-grained sample. However,
permeability from μ-CT flow modelling in the coarse-grained sample spanned more than two orders of
magnitude range that could point on the inhomogeneity of the rock on a larger (cm) scale.

For Sample S2 no flow modelling was possible because no REV has been found and the sample

demonstrates a poor pore network connectivity at the resolution scale. Gas permeability for this quartz wacke
layer S2 (Table 2) was about 2 orders of magnitude lower than that of the quartz arenite layers S1 and S3. The
low permeability regardless of the relatively high porosity in S2 (Table 2) is due to clay-rich matrix that
encloses a substantial void space (Hurst and Nadeau, 1995).
**5.    Conclusions**

Three consecutive sandstone layers of Hatira Formation of the Kurnub Group (Lower Cretaceous) from

northern Israel were comprehensively investigated by an integrated analytical program consisting of:
experimental petrographic and petrophysical methods, 3D μ-CT imaging and pore-scale flow modelling. The
following findings were obtained:



1.     All three sandstone layers show petrographic characteristics of the Kurnub Group in
the Levant. The main features are textural maturity (grain roundness, and sorting) and
mineralogical maturity (very small proportion of feldspars, kaolinite as the only clay mineral
detected by X-ray diffraction) suggesting a redeposition of Palaeozoic Nubian Sandstones. The
sedimentological features - cross bedding, graded bedding and interbedding of a horizon enriched
in silt and clay between the quartz arenite beds - may suggest a fluvial environment of deposition.
No fossils or carbonate components were detected.
2.     A higher extent of Fe-ox cementation was observed in the top quartz arenite
sandstone layer. Alternatively, a low cementation was observed in the bottom quartz arenite
sandstone layer located below the intermediate 20 cm thick impervious quartz wacke sandstone
layer. We suggest that the difference in the extents of cementation is related to the meteoric water
flux which supplied the iron solute, which was lower at the bottom sandstone layer below the
impervious intermediate layer.
3.     Two scales of porosity variations were found in the upper layer identified with
variogram analysis: fluctuations at 300 μm scale due to size variability of grains and pores, and
at 2 mm scale due to the appearance of high and low porosity bands. Local patches of grain
coating and meniscus type cementation were found related to locations of exceptional large
grains surrounded by regions with large pores, where preferential paths of fluid are more
plausible to flow through. These paths of infiltrated water supplied iron solutes to result in iron
oxide cementation at the adjacent regions with higher surface area, where the oxygen was
supplied by infiltration of air through the partly-saturated realm conditions. This cementation
pattern generated porosity fluctuations at ~2 mm scale.
4.     We suggest that in the bottom layer the changes of geochemical gradients at the
vicinity of the water table caused dissolution of $Fe^{3+}$ followed by re-precipitation of Fe-ox
across water level interface. Water level changes resulted in parallel banding interpreted as
"Liesegang bands".
5.     Sandstone colour was affected by the extent of cement and fines. The upper layer
with high cementation of Fe-ox was yellow-brown suggesting it is a goethite mineral, the bottom



layer with low cementation was pale red suggesting it is a hematite mineral, and the quartz wacke
sandstone was grey-white due to high extent of kaolinite mineral.
6.   Large-scale laboratory porosity and permeability measurements conducted in the
layers show lower variability for the quartz arenite (top and bottom) layers, and high variability
for the quartz wacke (intermediate) layer. These are confirmed also by anisotropy and
heterogeneity analyses conducted in the μ-CT-imaged geometry.
7.   Micro-scale geometrical rocks properties which were quantified in each layer (pore
size distribution, pore throat size, characteristic length, pore throat length of maximal
conductance, specific surface area, connectivity index, grain roughness) and macro-scale
petrophysical properties (porosity and tortuosity), along with conducted anisotropy analyses,
reflect the layers texture and differences between them. Combined, these characteristics explain
and qualify the permeability of the studies layers evaluated in our study by experimental and
computational methods.
8.   Macroscopic permeability upscaled from pore-scale velocity field simulated by free
flow modelling in real μ-CT-scanned geometry on mm-scale samples for the top and bottom
layers, showed agreement with lab petrophysical estimations on a cm-scale samples. Obtained
permeability anisotropy correlates with the presence of beddings. The scale including this kind
of anisotropy rather than a lower variability pore-scale, controls the macroscopic permeability.
Therefore, we suggest that in order to upscale reliably to the lab permeability at the scale of
permeameter, a sufficient large modelling domain is required to capture the textural features that
appear at the scale intermediate between the pore scale and lab permeameter scale.







## Appendix A

## Results of REV determination by the classical approach

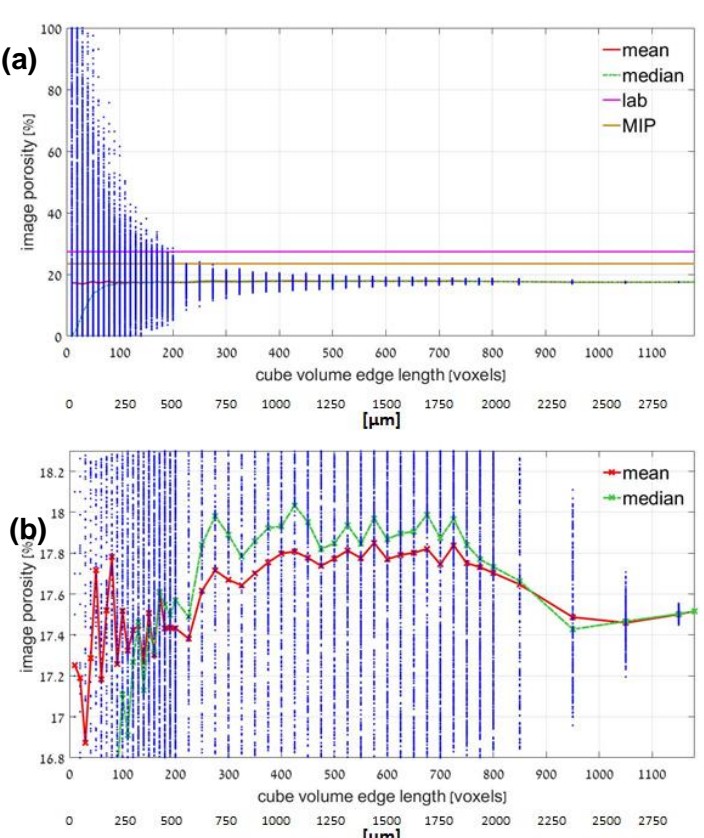

***Figure A1. (a)** Classical REV, Sample S1. The scattering of porosity measured for each sub-volume is shown in blue dots. Mean and median porosity were calculated for the varying edge size. Laboratory porosity measured by gas porosimeter is shown by a pink line. Image porosity for CT which was predicted by MIP for the resolution size is shown by yellow line. Mean and median porosity are depicted by red and green lines, respectively. **(b)** Zoom into the mean and median porosity trends. Mean and median curves converge starting from about 475 voxel-size of sub-volume (1187μm). Therefore, REV from the classical analysis is determined as a cube of 475 voxel (1187μm) edge length.*




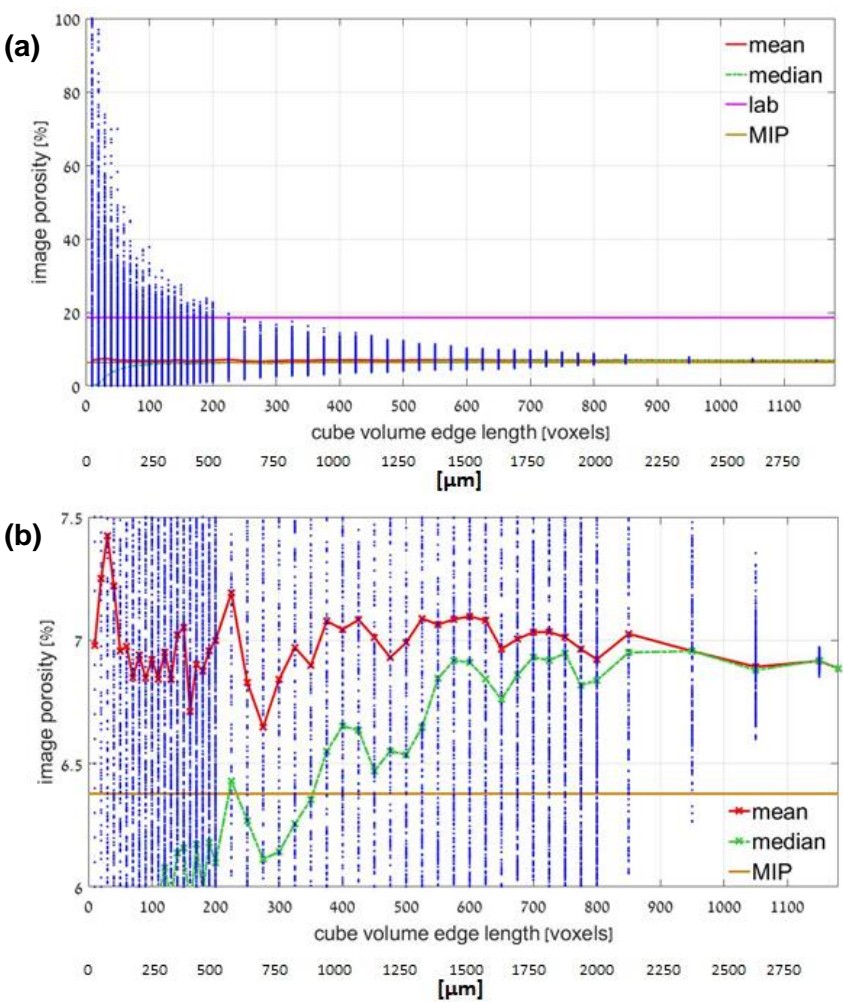

*Figure A2. (a) Classical REV, Sample S2. (b) Zoom into the mean and median porosity trends. Mean and median converge at 950 voxel (2375 μm) sub-volume size, which approaches the size of the entire sample (although the scattering remained high: 6.3% and 7.8% for min and max porosity, respectively). Therefore, no REV can be found by the classical REV approach.*



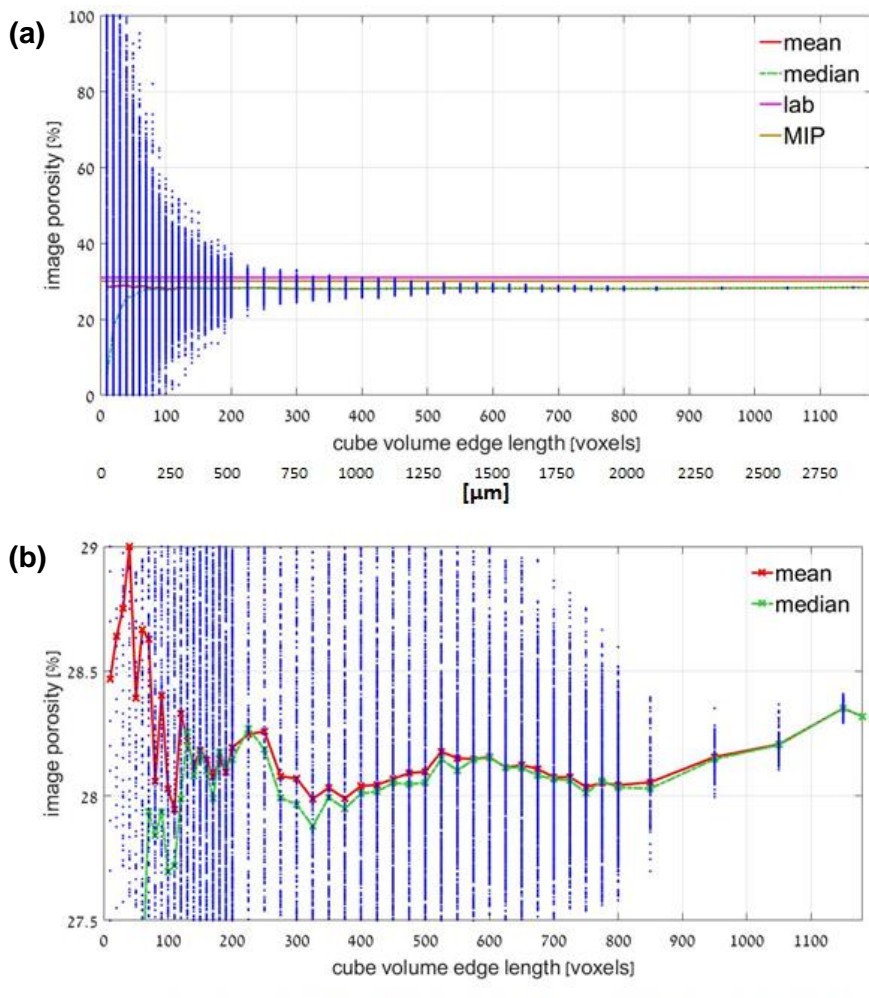

*Figure A3. (a) Classical REV, Sample S3. (b) Zoom into the mean and median porosity trends. Mean and median converge from ~350 voxel sub-volume edge length (875µm). Therefore, REV from the classical analysis is determined as a cube of 350 voxel (875µm) edge length.*

**Acknowledgments**

This project was supported by the fellowships from the Ministry of Energy, Israel, and from the University of Haifa. The authors are grateful to Igor Bogdanov for his continuing scientific support. Special thanks are to Rudy Swennen and his group from KU Leuven for contribution to MIP, thin sections preparations,



microscopy and μ-CT image processing; to Veerle Cnudde and her group from Ghent University for teaching
the image processing; and to Kirill Gerke and Timofey Sizonenko from Russian Academy of Sciences for
providing their code for image processing.

**Competing interests.** The authors declare that they have no conflict of interests.

**Author contributions.** PH and RK designed the study. PH developed codes on pore-scale modelling with
contributions by RK and MH. BS advised in microscopy and led the geological interpretations. MH scanned
the samples and contributed to statistical analysis conducted by PH. NW led the lab measurements. All co-
authors participated in analysis of the results. PH wrote the text with contributions from all co-authors. All co-
authors contributed to the discussion and approved the manuscript.

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
