# Peer review of "Petrographic and Petrophysical Characteristics of Lower Cretaceous Sandstones from northern Israel, determined by micro-CT imaging and analytical techniques"

_Solid Earth, 2019_

## Referee Comment (RC1) · Anonymous Referee #1 · 4 Mar 2019

The authors present a regional study on petrographic and petrophysical properties of a layered sandstone from Israel. The paper is not meant to improve process understanding but aims at characterizing the rock with a broad mix of methods (laboratory, imaging, simulation). Therefore, the value of this paper is supposedly to provide a reference for future studies working in this rock formation.

The paper is a bit lengthy, but easy to follow for its good language and clear structure. Some more data could potentially be outsourced into the appendix. I can only comment on the imaging and flow simulation parts of the paper, as I have very limited expertise

in geology. Hopefully, this is covered by other reviewers. The following comments address only minor details:

-line 200-2013: It is unclear to me how many simplifying assumptions are in the Katz & Thompson approach to derive lc, lmax and eventually k from MIP data. For instance, is the ratio 1/89 defined for a capillary bundle model with a specific shape of the cross section, or some percolation-type network model? Some more information should be provide here or later in the discussion section.

-line 229-240: What software was used for the non-local means filter, converging active contours, etc.? How were the parameters for each processing step determined? Manual by expert knowledge and then kept constant for all three samples? Software information is also missing for variogram analysis (line 262).

- line 303-305: Please explicitly state which software was used to determine tortuosity. Somewhere later in the text it was mentioned that Comsol was used (line 594).

- Line 308-313: How where the number of grains determined? By watersheding on the distance map of the binarized grain images?

- Line 350: Unclear, which method was used to capture Fig 3(i)

- line 667: Meaning of the sentence unclear to me: "A possible . . ."

- line 732: I'm well aware of the partial volume effect, but it is unclear how you can quantify surface roughness from the volume fraction of unresolved porosity. Please explain.

-line 744-759: Since the number of grains N is always positive, the connectivity index should always have the same sign as the Euler characteristic. I wonder why all CI values are positive in Table 2 are positive. A well connected pore network such as sample 3 should evoke a very negative Euler characteristic. Please explain in the text.

Line 911-912 and Line 930-931: So is it 84% or 50%? I guess one corresponds to S1

and the other to S3, but this needs to be made clear in the text.

- How many replicates plugs per layer and method? Information appears sporadically at several occasions in the manuscript (e.g. line 391, 423, 476 Figure 10). Could you add this information at an appropriate location in one of the tables?

What's the information gain between Fig. 7(a) and Fig. 8? Maybe merge both figures into one.

---

## Referee Comment (RC2) · Anonymous Referee #2 · 6 Mar 2019

Review of

Petrographic and Petrophysical Characteristics of Lower Cretaceous Sandstones from northern Israel, determined by micro-CT imaging and analytical techniques

Haruzi et al. present detailed descriptions and analyses of sandstones from northern Israel. They synthesize several analyses from the lab, including microtomography, and from fluid flow modelling. However, the motivation and resulting implications of their exhaustive characterizations are not at all clear from the abstract, introduction

or discussion. Presumably, they chose to focus on these rocks because they can be reservoir rocks. If this is the central motivation, then the abstract should state this point, and the discussion should describe how their analyses inform the potential productivity of these lower Cretaceous sandstones. Then, the organization of the paper should follow from this motivation and determine which figures remain in the main text, or are moved to the appendix. In its present form, the paper is very long, and so does not focus in sufficient detail on the significant contributions of this work. I recommend changing the organization of the paper to focus on conclusions #6-8, and keeping only the highlights from conclusions #1-5 in the main text. If as the abstract states, "core part of the study is the investigation of macroscopic permeability, upscaled from pore-scale velocity field" then the bulk of the main text should focus on presenting these results, rather than describing the potential depositional environment of each sample, for example.

Determining appropriate representative elemental volumes, and how to upscale porosity and permeability measurements in the lab to crustal scales are important questions that should be explored in more depth here. The general conclusion that larger models are needed to capture features at scales larger than the pore scale is obvious. The more relevant question is how large is "sufficiently large" (line 49)?

On a technical note, several sections of the paper seemed misplaced, including aspects of the results that are presently in the discussion. Some sections of the paper are written clearly, while others have serious grammatical errors, such as sentences and phrases that lack verbs. In several places there are bolded phrases that I suppose the authors intend to be headings, but they lack the typical notation for sections, i.e., 1.1.

I would not recommend publishing this work in its present form. I suggest reorganizing this work after deciding on the main motivation of the study, and focusing in detail on the points that answer this motivation. The comparing the significant contributions of this work to previous studies on sandstone reservoirs would also help this paper have

relevance.

I list more detailed points below:

1. Abstract: Motivate studying the characteristics in abstract. As mentioned above, be clear about the central motivation for this work.

2. Section 1: Rearrange the sections to make 1.1 only part of intro, and make 1.2 background as new section 2.

3. Line 107: overly should probably be overlie

4. Methods: Remove list with roman numerals, organize into true sections that align with the journals' format.

5. Line 272, line 632, and probably several other places: parentheses should be put around (Fig. X)

6. Line 308: Describe what the Euler characteristic shows

7. Line 314: There is a strange green box around a bullet point.

8. Section 4.1-4.2 should be in the results, not the discussion

9. Line 678: "post_depositional " seems to have an underscore

10. Line 833: rewrite with verb "mean porosity lower than the median one "

11. Paragraph at line 841: rewrite this paragraph. There are many grammatical errors

12. Line 899: "In contrast the," change position of comma

13. In the discussion, it would be useful to compare your porosity and permeability measurements to measurements from similar potential reservoir rocks. This comparison will help make this work relevant to the broader community.

---

## Author Comment (AC1) · 7 May 2019

**Authors response on comments from anonymous Referee #1:**

The authors would like to thank both anonymous referees for their comments and suggestions. We would like to comment on them point by point as recommended. Additionally, we would like to give a general outlook on the changes, which are implemented in the revised version of the manuscript.

**General comment:**

The authors present a regional study on petrographic and petrophysical properties of a layered sandstone from Israel. The paper is not meant to improve process understanding but aims at characterizing the rock with a broad mix of methods (laboratory, imaging, simulation). Therefore, the value of this paper is supposedly to provide a reference for future studies working in this rock formation.

The paper is a bit lengthy, but easy to follow for its good language and clear structure. Some more data could potentially be outsourced into the appendix. I can only comment on the imaging and flow simulation parts of the paper, as I have very limited expertise in geology. Hopefully, this is covered by other reviewers.

**Authors' statement:**

The main goal of the paper is to provide a comprehensive and multi-methodological case study on these particular Lower Cretaceous sandstones from the north of Israel as a fundamental base for future works. This goal is now stated explicitly in the abstract and in the introduction.

The implemented multi-methodological multi-scale approach allows also a better process understanding. Nevertheless, since we recognize that this aim and scope of the paper has not been fully presented as intended in the initial version of the manuscript, we are going to slightly change the title of the manuscript, clarifying that this is a fundamental case study ("Multi-methodological Petrographic and Petrophysical Case Study of Lower Cretaceous Sandstones from Hatira Formation, northern Israel"). Additionally, we shortened and re-organized the manuscript as described in our responses to both reviewers, in order to make it more accessible and informative for the potential reader of Solid Earth.

**Detailed remarks:**

*Comment on line 200-213:* It is unclear to me how many simplifying assumptions are in the Katz & Thompson approach to derive lc, lmax and eventually k from MIP data. For instance, is the ratio 1/89 defined for a capillary bundle model with a specific shape of the cross section, or some percolation-type network model? Some more information should be provide here or later in the discussion section.

*Response:* We added the following information regarding the approach in the Appendix. Katz and Thompson (1986, 1987) developed a permeability model derived from the percolation theory (Ambegaokar et al., 1971). The model is applicable for systems characterized by a broad distribution of local conductances with only short-range correlations, like those that occur in sandstone with broad range of size distribution of pore spaces. The constant $\frac{1}{89}$ resulted from a trial solution.

*Comment on line 229-240:* What software was used for the non-local means filter, converging active contours, etc.? How were the parameters for each processing step determined? Manual by expert knowledge and then kept constant for all three samples? Software information is also missing for variogram analysis (line 262).

*Response:* The code for Image Processing was designed by Kirill Gerke (in Acknowledgements) and received personally from him. The parameters were determined manually for each sample by expert knowledge, in order to derive the best possible results for the variety of sample material. Variogram analysis was performed using 'Variogramfit' Matlab package. This information appears now in the Methods Sect.

*Comment on line 303-305:* Please explicitly state which software was used to determine tortuosity. Somewhere later in the text it was mentioned that Comsol was used (line 594).

*Response:* 1) particle tracing after Stokes flow simulation was implemented with Comsol Multiphysics software (specified initially in lines 292-293 of the manuscript); 2) shortest path simulation through the main pore network with Fast Marching Method (Sethian, 1996, former lines 303-305) was implemented with Matlab using Accurate Fast Marching plug-in. We have added the names and references to all of used softwares within the Methods section.

*Comment on line 308-313:* How where the number of grains determined? By watersheding on the distance map of the binarized grain images?

*Response:* Separation of grains from the sample cluster was performed using distance map followed by watershed algorithms (MorphoLibJ plugin, Legland et al., 2014). This clarification is inserted in the manuscript at the introduction of the image analysis.

*Comment on line 350:* Unclear, which method was used to capture Fig 3(i)

*Response:* Incident light microscopy has been used for Figure 3(i). We have added this information.

*Comment on line 667:* Meaning of the sentence unclear to me: "A possible : : :"

*Response:* This sentence is now re-phrased: A possible source for the clay (0.8 %) is pressure solution (Fig. 5d).

*Comment on line 732:* I'm well aware of the partial volume effect, but it is unclear how you can quantify surface roughness from the volume fraction of unresolved porosity. Please explain.

*Response:* We withdrew the discussed paragraph.

*Comment on line 744-759:* Since the number of grains N is always positive, the connectivity index should always have the same sign as the Euler characteristic. I wonder why all CI values are positive in Table 2 are positive. A well-connected pore network such as sample 3 should evoke a very negative Euler characteristic. Please explain in the text.

*Response:* The reviewer is correct regarding the sign of Euler characteristic. $\chi$ was modified to $|\chi|$ in Eq.7. Please see the additional detail in our response to the comment #6 of Referee 2.

*Comment on line 911-912 and 930-931:* So is it 84% or 50%? I guess one corresponds to S1 and the other to S3, but this needs to be made clear in the text.

*Response:* The reviewer is correct. 50% corresponds to S1 while 84% to S3. This has been clarified in the text.

*Comment on "number of plugs per layer and method":* How many replicates plugs per layer and method? Information appears sporadically at several occasions in the manuscript (e.g. line 391, 423, 476 Figure 10). Could you add this information at an appropriate location in one of the tables?

*Response:* This information has been added in Table 2.

*Comment on Fig. 7(a) and Fig. 8:* What's the information gain between Fig. 7(a) and Fig. 8? Maybe merge both figures into one.

*Response:* Figures 7a and 8 were merged to one figure. 7b was removed.

---

## Author Comment (AC2) · 7 May 2019

**Authors response on comments from anonymous Referee #2:**

The authors would like to thank both anonymous referees for their comments and suggestions. We would like to comment on them point by point as recommended. Additionally, we would like to give a general outlook on the changes, which were implemented in the revised version of the manuscript.

**General comment:**

However, the motivation and resulting implications of their exhaustive characterizations are not at all clear from the abstract, introduction or discussion. Presumably, they chose to focus on these rocks because they can be reservoir rocks. If this is the central motivation, then the abstract should state this point, and the discussion should describe how their analyses inform the potential productivity of these lower Cretaceous sandstones**....** I recommend changing the organization of the paper to focus on conclusions #6-8, and keeping only the highlights from conclusions #1-5 in the main text. If as the abstract states, "core part of the study is the investigation of macroscopic permeability, upscaled from porescale velocity field" then the bulk of the main text should focus on presenting these results, rather than describing the potential depositional environment of each sample, for example. Determining appropriate representative elemental volumes, and how to upscale porosity and permeability measurements in the lab to crustal scales are important questions that should be explored in more depth here.

The comparing the significant contributions of this work to previous studies on sandstone reservoirs would also help this paper have relevance.

**Authors' statement:**

The main goal of the paper is to provide a comprehensive and multi-methodological petrographical and petrophysical case study on these particular Lower Cretaceous sandstones from the north of Israel as a fundamental base for future works. This goal is now stated explicitly in the abstract and in the introduction.

However, there is no statement in the manuscript that the present study is being about reservoir rocks. Thus, the suggestion of Referee 2 ("Presumably, they chose to focus on these rocks because they can be reservoir rocks") is not appropriate. We do not intend to focus on permeability only and thus to analyse "the potential productivity of these lower Cretaceous sandstones" as the Referee suggested, also because there is no indication whether they would serve as reservoirs rocks in the northern Israel.

Instead, it is stated in the corrected abstract and introduction that "An applied multi-methodological multi-scale approach allows also a better process understanding and an identification of connection between the parameters evaluated at the different scales". The samples were chosen for investigation because of their differences immediately observed at the various scales: at the outcrop, in the hand specimens, and under a binocular microscope.

Nevertheless, since we recognize that this aim and scope of the paper has not been fully presented as intended in the initial version of the manuscript, we are going to slightly change the title of the manuscript, clarifying that this is a fundamental case study ("Multi-methodological Petrographic and Petrophysical Case Study of Lower Cretaceous Sandstones from Hatira Formation, northern Israel"). Moreover, we shortened and re-organized the manuscript, as described below, in order to make it clearer and more informative for the potential reader of Solid Earth.

The main changes incorporated in the manuscript are listed below:

- The Abstract is shortened and reformulated according to the aim and scope of the paper (as stated above)
- The Introduction was changed to clarify the significance of the study (as stated above)
- The Geological setting (Sect.2 now) is shortened
- The Methods section is significantly shorter and reorganized, in accordance with the format of the journal, and following Comment 4 (below).
- The following changes were implemented in the Results Sect.:

  -The text is shortened

  -Verbal description and figures of classical REV are moved to the Appendix

  -In Fig.3 parts f-h are removed,

  -Figs 7-8 are merged, Fig.9 is moved to the Appendix.

  -Description of the classical REV is removed, while the relevant pictures from the Appendix are referenced in the text.

  -A new correlation plot between porosity and clay content is added to the former Fig.11.

- The Discussion is now composed of three subsections:

  5.1 Influence of microscopic pore network characteristics on permeability. This addresses a "process understanding" specified as one of the main objectives of our paper.

  5.2 Upscaling permeability: accuracy of the extended computational workflow. It discussed the methodological aspects of our study.

  5.3 Inferences on the sedimentology and diagenesis of the samples in the context of the Hatira Formation. It suggests the insights that predefine the similarity and differences between the studied layers.

- The Conclusions are now focused more on former points 6-8.

**Detailed remarks:**

*Comment on line 49:* "how large is "sufficiently large"

We have re-phrased this sentence in the text to clarify this point (and removed it from the abstract). Specifically, textural bedding within a 2 mm scale dominates the flow anisotropy. Implicitly, a "sufficiently large" modelling domain would be "as large as necessary" bounded by the scale of this feature (i.e. > 2 mm edge length) and – additionally – "not larger than necessary" (i.e. << 10 mm) in order to optimize the computational efforts in a sufficient amount of time. Now, this point is specified in the Discussion.

*Comment on entire manuscript:*

1. Comment 1: Abstract: Motivate studying the characteristics in abstract. As mentioned above, be clear about the central motivation for this work.

Response: The abstract was modified and shortened based on the main goal of the paper defined in the general remarks above.

2.Comment 2: Section 1: Rearrange the sections to make 1.1 only part of intro, and make 1.2

background as new section 2.

Response: Performed.

*3. Comment on Line 107:* overly should probably be overlie

Response:  Correct, has been changed.

4. *Comment on Methods*: Remove list with roman numerals, organize into true sections that align with the journals' format.

*Response:* Text was reorganized and shortened according to the journal format.

5. *Comment:* Line 272, line 632, and probably several other places: parentheses should be put around (Fig. X)

*Response:* Implemented throughout the entire manuscript.

6. *Comments on line 308:* Describe what the Euler characteristic shows

*Response:* We defined the Euler characteristic in the Methods Sect. and put the following information in the Appendix:

Euler characteristic is a number that describes the structure of a topological space. The most intuitive way to think about the Euler characteristic is in terms of its Betti numbers ($\beta_i$).

$$\chi = \beta_0 - \beta_1 + \beta_2$$

For a 3D object, $\beta_0$ is the number of components, $\beta_1$ is the number of inequivalent loops and $\beta_2$ is the number of cavities (enclosed voids). In describing the topology of the pore space of a porous rock, it can be assumed that the solid matrix is connected, so that $\beta_2 = 0$. In this case, the Euler number reduces to the difference between the number of discrete components and inequivalent loops. If all pore space is connected via one pathway or another, and assuming that there are no isolated pore spaces, then $\beta_0 = 1$. In a pore network of sandstone which can be modeled as a bundle of tubes, the number of loops $\beta_1$ is large and $\chi$ is negative. Therefore, Euler number is related to the connectivity of the pore space. As the number of loops decreases, the Euler number becomes less negative and will eventually become positive, at which point the system will no longer percolate, according to Vogel (2002).

H.-J. Vogel, Topological Characterization of Porous Media, in Morphology of Condensed Matter, K. Mecke and D. Stoyan, Editors. 2002, Springer Berlin Heidelberg. p. 75-92.

7. *Comment in* Line 314: There is a strange green box around a bullet point.

*Response*: Removed.

8. *Comment:* Section 4.1-4.2 should be in the results, not the discussion

*Response:* Please see the description of the modified discussion above in the response to the General Remarks, and its motivation.

9. *Comments on line 678:* Line 678: "post_depositional " seems to have an underscore

*Response:* Correct, has been removed.

10. *Comments on line 833:* rewrite with verb "mean porosity lower than the median one "

*Response:* Has been re-phrased.

11. *Comments in line 841 ff.:* rewrite this paragraph. There are many grammatical errors.

*Response:* The entire paper has been proof-read and re-phrased where needed, including the addressed paragraph.

12. *Comments in line 899:* "In contrast the," change position of comma

*Response*: Has been changed.

13. *Comments on discussion section:* In the discussion, it would be useful to compare your porosity and permeability measurements to measurements from similar potential reservoir rocks. This comparison will help make this work relevant to the broader community.

*Response*: The comparison was presented even in the initial version of the manuscript (e.g. lines 736-742, 920-925, 941-943). We added some data in the corrected version of the paper as well.